# Neuron-based Personality Trait Induction in Large Language Models

**Jia Deng**[1], **Tianyi Tang**[2], **Yanbin Yin**[1], **Wenhao Yang**[3], **Wayne Xin Zhao**[1,*] **Ji-Rong Wen**[1]

[1]Gaoling School of Artificial Intelligence, Renmin University of China.
[2]Tongyi Lab.
[3]Institute of Statistics and Big Data, Renmin University of China.
`dengjia0510@outlook.com, steventianyitang@outlook.com`
`batmanfly@gmail.com, jrwen@ruc.edu.cn`

## Abstract

Large language models (LLMs) have become increasingly proficient at simulating various personality traits, an important capability for supporting related applications (*e.g.,* role-playing). To further improve this capacity, in this paper, we present a neuron-based approach for personality trait induction in LLMs, with three major technical contributions. First, we construct PERSONALITYBENCH, a large-scale dataset for identifying and evaluating personality traits in LLMs. This dataset is grounded in the Big Five personality traits from psychology and is designed to assess the generative capabilities of LLMs towards specific personality traits. Second, by leveraging PERSONALITYBENCH, we propose an efficient method for identifying personality-related neurons within LLMs by examining the opposite aspects of a given trait. Third, we develop a simple yet effective induction method that manipulates the values of these identified personality-related neurons. This method enables fine-grained control over the traits exhibited by LLMs without training and modifying model parameters. Extensive experiments validate the efficacy of our neuron identification and trait induction methods. Notably, our approach achieves comparable performance as fine-tuned models, offering a more efficient and flexible solution for personality trait induction in LLMs. We provide access to all the mentioned resources at https://github.com/RUCAIBox/NPTI.

## 1 Introduction

Recently, the potential of large language models (LLMs) has been widely explored, not only in generating human-like text but also in simulating various personality traits. Such capabilities are crucial for applications like role-playing (Pan & Zeng, 2023), gaming (Carlander et al., 2024), and therapeutic settings (Xu et al., 2023b), where nuanced personalities should be stimulated and established by the supporting system. Understanding and regulating the mechanism of possessing personality traits in LLMs is essential for developing responsive and adaptable AI systems.

To induce the personality traits in LLMs, existing research primarily adopt either prompt-based or training-based methods. Prompt-based methods (Tan et al., 2024; Huang et al., 2023; La Cava et al., 2024; Jiang et al., 2023b; Kovač et al., 2023) are efficient and can quickly induce personality traits without extensive retraining, while their performance highly depends on the prompt design and the foundation model. As a comparison, training approaches (Pan & Zeng, 2023; Liu et al., 2024; Li et al., 2022) provide greater stability but often require substantial time, computational resources, and high-quality datasets. To reduce the training costs, some study (Zhu et al., 2024) proposes to identify relevant attention heads and guide them in a specific direction to align with a particular personality. However, these studies either lack theoretical guidance (*e.g.,* psychological clues) in methodology or cannot impose fine-grained, precise control on nuanced personalities.

To address these issues, in this work, we present a **N**euron-based approach for **P**ersonality **T**raits **I**nduction in LLMs, named as **NPTI**. As the theoretical guidance, our approach is developed based

---

*Corresponding author

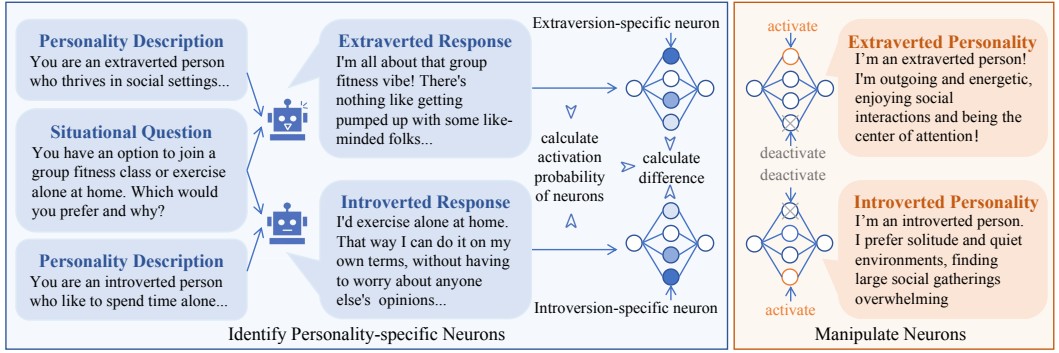

Figure 1: The overall workflow of our proposed approach NPTI. The left diagram first illustrates how to induce opposite aspects of the same personality trait (*e.g.,* extroversion and introversion) through prompts to address situational questions from PERSONALITYBENCH, while calculating the activation probabilities of neurons. We then calculate the differences in these probabilities between opposing responses to identify the neurons governing specific personality dimensions. Further, the right diagram illustrates how to activate neurons associated with one aspect while deactivating those associated to the opposing trait, thereby effectively altering the model's personality.

on the Big Five personality traits from psychology (Tupes & Christal, 1992), which categorizes personality into five traits: openness, conscientiousness, extroversion, agreeableness, and neuroticism. Following the Big Five personality traits, we construct a personality dataset PERSONALITYBENCH with specially curated instances (*i.e.,* a personality description paired with a situational question). Unlike existing related datasets mainly in the multi-choice format (Jiang et al., 2024), PERSONALITYBENCH conducts generative personality evaluations with real-world test cases for LLMs, which can effectively enhance the evaluation robustness. Subsequently, we employ PERSONALITYBENCH to identify personality-related neurons. Instead of simply measuring the activation degree (Zhu et al., 2024), we find that traits often correspond to two opposite aspects (*e.g.,* extroverted and introverted), and thus propose to calculate the activation difference of a LLM when examining the opposing aspects of a given trait. Furthermore, to impose precise control on personality traits, we design a simple yet effective induction method by manipulating the values of the personality-related neurons. We seek *neurons* as the manipulation units as it directly conveys fine-grained semantics on personality traits, and existing studies (Dai et al., 2021; Meng et al., 2022; Tang et al., 2024) have shown that neuron manipulation can effectively elicit or alter the behaviors of LLMs.

To summarize, this work presents a systematic personality trait induction approach, with the main technical contributions in three aspects:

• *Personality dataset*: We design a dataset named PERSONALITYBENCH, comprising 180,000 open-ended questions specifically crafted for each of the Big Five personality traits, in which the answers clearly distinguish between high and low levels of these traits in the model.

• *Neuron identification*: We propose a new identification method for locating the neurons associated with specific personality traits in LLMs by leveraging both the positive and negative aspects of the same personality trait.

• *Neuron manipulation*: We design a simple yet effective manipulation method for personality-related neurons, with specific modification strategies for opposite aspects of a given trait. Extensive experiments using various evaluation methods on different LLMs have verified the effectiveness and generality of our method.

## 2 RELATED WORK

**Personality in LLMs.** As large language models advance (Zhao et al., 2023; Hu et al., 2024), their improved human-like text interactions are increasingly used for simulating diverse personalities (Wen et al., 2024). This capability enables language models to perform diverse functions, such as conducting sociological experiments (Ziems et al., 2024; Park et al., 2023) and impersonating spe-

cific characters (Li et al., 2023; Shao et al., 2023; Wang et al., 2024b). Current research on personality in LLMs primarily falls into two categories: *personality assessment* and *personality induction*. Researchers commonly utilize LLMs to assess the personalities of existing LLMs. One approach involves directly utilizing LLMs to respond to questionnaires based on established personality frameworks such as the Big Five personality traits (Tupes & Christal, 1992) or the Myers-Briggs Type Indicator (MBTI) (Boyle, 1995). Alternatively, LLMs can be used to infer personality traits from given texts. This can be done by analyzing the text itself to make assessments (Peters et al., 2024; Ji et al., 2023), or by incorporating LLMs to enhance existing personality recognition models (Cao & Kosinski, 2024). As for personality induction, one approach to achieve it is through *prompt-based induction* (Tan et al., 2024; Huang et al., 2023; La Cava et al., 2024; Jiang et al., 2023b; Kovač et al., 2023), which can be further divided into explicit prompting (Xu et al., 2023a) and implicit prompting (Suzgun et al., 2022). Explicit prompting gives direct descriptions of personality traits for the model, while implicit prompting provides real-world examples, letting the model infer traits via in-context learning. For example, Jiang et al. (2024) use explicit personality prompts to guide LLMs in generating responses tailored to specific personality traits in open-ended questions (Kwantes et al., 2016). The other approach to achieving personality induction is through *training*, such as pre-training (Pan & Zeng, 2023) and post-training (Liu et al., 2024). Li et al. (2022) employ direct preference optimization to fine-tune Llama-2-chat-7B with question-answer pairs from the Big Five Inventory (BFI) (John et al., 1999), finding that this approach significantly enhanced the model's performance on the Short Dark Triad (Paulhus & Williams, 2002). Meanwhile, Zhu et al. (2024) train "probes" to capture "activations" of each attention head, helping aligning individual preference characteristics during the model's forward computation. However, prompt-based induction can produce varying and unstable results depending on the specific prompts used. Model training, while more consistent, requires significant time and computational resources and often suffers from a lack of labeled data. In contrast, our neuron-change method is more lightweight and stable, enabling models to exhibit specific personalities without the need for extensive training.

**Knowledge Mechanisms in LLMs.** Research has revealed that *neurons* in neural networks play a role in storing the knowledge acquired during training (Mu & Andreas, 2020; Bau et al., 2020; Geva et al., 2022). As a widely adopted architecture for language models, the Transformer (Vaswani, 2017) is composed of multiple layers featuring self-attention mechanisms and feed-forward neural networks (FFNs). Early studies demonstrate that attention heads in pre-trained models, such as BERT (Devlin, 2018), can capture and convey various forms of knowledge within the model (Voita et al., 2019; Clark, 2019; Hoover et al., 2019). With the development of large language models, researchers increasingly focus on exploring and exploiting the knowledge mechanisms (Wang et al., 2024a) embedded within language models. Olsson et al. (2022) present evidence suggesting that a combination of several attention heads can exhibit pattern copying behavior and be responsible for the universal in-context learning capabilities. Meanwhile, several studies explore how to leverage the memory of neurons inside FFNs to modify the model's behavior, such as editing specific factual knowledge (Dai et al., 2021; Meng et al., 2022) and changing the output language of LLMs (Tang et al., 2024). Besides instance-level editing, the FFNs in the last few layers show greater importance when editing conceptual knowledge according to Wang et al. (2024c). Unlike previous research, our work introduces a novel approach that identifies neurons within FFNs responsible for controlling personality traits and modifies them to exhibit specific personality characteristics.

## 3 APPROACH

In this section, we introduce the proposed personality induction method, **NPTI**, for LLMs. Our approach is built upon a meticulously curated dataset called PERSONALITYBENCH, which is detailed in Section 3.1. It comprises two main steps: identifying personality-related neurons (Section 3.2) and manipulating these neurons to induce the desired personality traits in LLMs (Section 3.3). The overall procedure of our approach is depicted in Figure 1.

### 3.1 PERSONALITYBENCH CONSTRUCTION

We first describe the construction process of our generative benchmark, PERSONALITYBENCH, designed to evaluate the ability of LLMs to exhibit consistent personality traits. This benchmark also facilitates our identification of personality-related neurons within LLMs. Unlike traditional

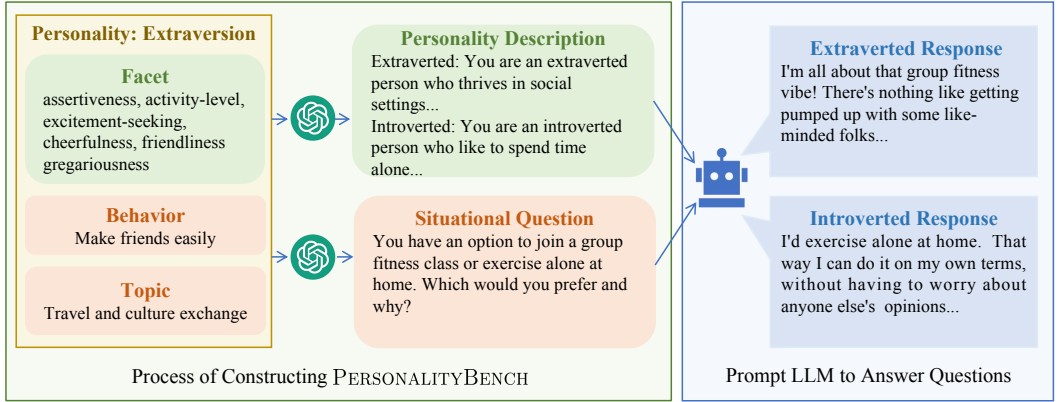

Figure 2: Flowchart for constructing PERSONALITYBENCH.

multiple-choice-based approaches like BFI (Tupes & Christal, 1992) and MBTI (Boyle, 1995), our benchmark conducts the test in the form of generative tasks. Since multiple-choice questions may lead to evaluation bias or even inaccuracies (Dorner et al., 2023), they can't well evaluate the ability of LLMs to generate natural responses that consistently reflect specific personality traits in real-world scenarios. We denote the constructed PERSONALITYBENCH as $\mathcal{P}$, each instance within $\mathcal{P}$ consists of a *personality description* paired with a *situational question*. Next, we introduce the approach to constructing the personality description and situational question in detail.

**Personality Description Generation.** We generate personality descriptions based on "Big Five personality traits" (Tupes & Christal, 1992), which categorizes personality into five major traits: *openness* (O), *conscientiousness* (C), *extroversion* (E), *agreeableness* (A), and *neuroticism* (N). Following this, each instance in our PERSONALITYBENCH $\mathcal{P}$ is associated with a given personality trait $t$, where $t$ belongs to the Big Five traits: $\{O, C, E, A, N\}$. The subset of the instances with the personality trait $t$ is denoted by $\mathcal{P}_t$. Furthermore, as described by McCrae & John (1992), each trait can be further broken down into more detailed facets. For example, the trait of openness corresponds to imagination, artistic interests, emotionality, adventurousness, intellect, and liberalism. These facets can be utilized to extend personality descriptions, generating diverse examples following Jiang et al. (2024). Specifically, we employ ChatGPT to create concise descriptive sentences in the second person by extending the adjective forms of these traits. For traits with opposite aspect (*e.g.,* introverted), we incorporate their antonyms. Initially, we produce a description for each of the ten personality aspects, then refine these manually. These refined descriptions serve as exemplars for generating more diverse descriptions.

**Situational Question Construction.** Using the generated personality descriptions, we design situational questions aimed at eliciting distinct responses across personality traits. Traditional evaluation questions (Hilliard et al., 2024), such as "*What do you usually do at the weekend?*" often fail to capture meaningful differences across personality types, as they may prompt superficially similar responses. To address this, we design a set of targeted questions grounded in real-world behaviors, which are tailored to amplify personality-related behaviors. Specifically, we utilize the IPIP-NEO-300 questionnaire (Goldberg et al., 1999; 2006) to generate the situational questions that reflect various real-world behaviors. This questionnaire provides a detailed investigation into individuals' behaviors across different personality facets. For instance, an "*adventurous*" individual is characterized by a tendency to "*likes to visit new place*". To further diversify the questions, we incorporate common real-world topics introduced in UltraChat (Ding et al., 2023), such as technology, environment, and arts. To generate the situational questions, we employ specially designed prompts for ChatGPT to simulate complex scenarios involving dilemmas, conflicting priorities, or challenging decisions that align with specific facets based on specific behaviors and topics. We further refine the results with ChatGPT to review for potential bias–whether moral or emotional–and make necessary improvements. This refinement ensures that the questions can better capture the relevant personality traits while maintaining objectivity. Detailed prompts can be found in Table 14 and Table 15.

Finally, our PERSONALITYBENCH consists of 180,000 instances of the Big Five personality traits for neuron identification, with 36,000 instances for each trait on average. As for evaluation, we utilize a similar idea to construct situational questions based on SOCIALIQA (Sap et al., 2019), which has approximately 90 questions for each trait. We further conduct human evaluation to verify the quality of our datasets, which are provided in Appendix B.

## 3.2 IDENTIFYING PERSONALITY-RELATED NEURONS

Based on the PERSONALITYBENCH, we can identify neurons that regulate specific personality traits. In what follows, we first specify the meaning of neuron in our work, then present the identification method based on activation difference.

**Neuron Specification.** Currently, most LLMs are built upon an auto-regressive Transformer architecture (Vaswani, 2017), where the key components are multi-head self-attention (MHA) and feed-forward networks (FFN). Prominent LLMs, such as LLaMA (Touvron et al., 2023) and Gemma (Team et al., 2024), commonly employ GLU (Shazeer, 2020) as a variant of the activation function in the FFN module. Within a given layer, the FFN module can be expressed as:

$$\boldsymbol{h} = \left( \sigma \left( \hat{\boldsymbol{h}} \boldsymbol{W}_1 \right) \odot \left( \hat{\boldsymbol{h}} \boldsymbol{W}_3 \right) \right) \cdot \boldsymbol{W}_2, \tag{1}$$

where $\hat{\boldsymbol{h}} \in \mathbb{R}^d$ represents the output of the MHA module for a specific token in this layer. The function $\sigma(\cdot)$ typically denotes a non-linear activation function, such as SiLU (Ramachandran et al., 2017). The learned projection matrices are $\boldsymbol{W}_1 \in \mathbb{R}^{d \times d'}$, $\boldsymbol{W}_2 \in \mathbb{R}^{d' \times d}$, and $\boldsymbol{W}_3 \in \mathbb{R}^{d \times d'}$. In this context, a *neuron* is conceptualized as applying a linear transformation to a specific column of the weight matrix $\boldsymbol{W}_1$ followed by a non-linear activation function to the result.

**Activation-based Identification.** Given the subset $\mathcal{P}_t$ corresponding to a trait $t$ in PERSONALITYBENCH, we prompt LLMs with these instances to generate responses to questions. During this process, we calculate the activation probability of the $i$-th neuron in each layer when tokens are generated as follows:

$$\mathrm{Pr}_i = \frac{1}{n} \sum_{j=1}^{n} \mathbb{I} \left( \sigma(\hat{\boldsymbol{h}} \boldsymbol{W}_1)_i > 0 \right), \tag{2}$$

where $n$ is the total number of generated tokens and $\mathbb{I}$ represents the indicator function. Since one trait has positive ($t+$) and negative ($t-$) aspects (*e.g.,* extroverted and introverted), we can further compute the activation difference $\delta$ for the $i$-th neuron between the opposing personality traits:

$$\delta = \mathrm{Pr}_i^{t+} - \mathrm{Pr}_i^{t-}. \tag{3}$$

Finally, we set a difference threshold to identify personality-related neurons. Neurons are classified as controlling the positive aspect of trait $t$, denoted as $\mathbb{P}_t^+$, if $\delta$ exceeds 10%. Conversely, neurons with a $\delta$ below -10% are designated as controlling the negative aspect of trait $t$, represented by $\mathbb{P}_t^-$. This classification allows us to distinguish neurons that significantly influence specific personality traits in either a positive or negative direction.

## 3.3 MANIPULATING PERSONALITY-RELATED NEURONS

As the personality-related neurons largely influence the personality behaviors of LLMs, we can induce the LLMs' personality by adjusting the values of these neurons. To account for the varying importance of neurons with different $\delta$ values, we introduce a weighted function $f$ based on the Sigmoid function. This function assigns higher $f(\delta)$ values to neurons with larger $\delta$ values, reflecting their greater influence on personality traits. Our approach to eliciting a positive shift in personality trait $t$ involves increasing the activation value of positive neurons, deactivating negative neurons, and maintaining the original values of neutral neurons. The modified values for each neuron can be

formulated as follows:

$$n = \begin{cases} \min(0, n_{ori}), & \text{if neuron} \in \mathbb{P}_t^- \\ n_{ori} + \gamma \cdot a_{95} \cdot f(\delta), & \text{if neuron} \in \mathbb{P}_t^+ \\ n_{ori}, & \text{others} \end{cases} \tag{4}$$

where $n_{ori}$ represents the original neuron value, $\gamma$ is a hyperparameter controlling the magnitude of change, and $a_{95}$ denotes the 95th percentile of the neuron's original activation, which ensures the modification respects each neuron's upper bounds. For positive neurons, we aim to amplify their influence to steer the LLM towards a more positive trait expression. Conversely, we manually deactivate negative neurons to suppress their contribution to negative trait, inspired by Tang et al. (2024) that deactivating neurons tied to a specific language significantly weakens the model's output in that language, with minimal impact on others. To induct a negative shift in the personality trait, we reverse the conditions in Equation 4, deactivating positive neurons and enhancing negative ones.

## 4 EXPERIMENT

### 4.1 EXPERIMENTAL SETUP

**Models.**    We primarily conduct our experiments on the LLaMA-3-8B-Instruct model (Dubey et al., 2024), known for its impressive performance in natural language understanding and generation. It has strong capabilities and adapts well to various tasks, making it an ideal base model for our studies. To extensively assess the effectiveness of our method, we also incorporate other LLMs under various configurations: Mistral-7B-Instruct-v0.3 (Jiang et al., 2023a), Gemma2-9B-it (Team et al., 2024), and Qwen2.5-7B-Instruct (Yang et al., 2024) to verify the compatibility of our methods.

**Baseline Methods.**    We select the following five methods as our baselines:

- **Simple prompt induction**: This method employs a single adjective to guide the model toward different personality traits (*e.g., you are an "extraverted/introverted" person*). More specific adjectives and prompt can be respectively found in Table 13 and Table 16;
- $P^2$ **induction** (Jiang et al., 2024): In this approach, the model receives a detailed ChatGPT-generated description of a particular personality trait. Prompt is shown in Table 17;
- **PAS** (Zhu et al., 2024): This method involves using the IPIP-NEO-300 questionnaire to train a probe that identifies the attention heads most closely related to a specific personality trait. During testing, this probe is then used to adjust the model's personality;
- **ActAdd** (Turner et al., 2023): This method modifies the residual stream values of a single layer to induce model behavior during the output stage, using opposing prompts to determine the extent of the modifications;
- **Supervised fine-tuning (SFT)**: We employ LoRA training (Hu et al., 2021) to embody a specific personality. During training, we set the learning rate to 1e-4 with a cosine decay. The rank of LoRA is set to 8, and the batch size is configured to 8. Notably, this approach can be considered the upper limit of our method. The prompt we used during testing in presented in Table 18.

**Implementation Details.**    During the construction of our PERSONALITYBENCH benchmark, we employ `gpt-4o-20240806` API with greedy search. Our PERSONALITYBENCH has 180,000 instances for identifying neurons and around 450 instances for evaluating LLMs' personality induction. When identifying neurons of each LLM with PERSONALITYBENCH, we employ greedy search with a repetition penalty of 1.1 to answer situational questions. The number of identified neurons for each neuron is around 20,000 in LLaMA-3-8B. As for the hyperparamters in Equation 4, we set $\gamma = 1.4$ and assign $f(\delta) = \frac{1}{1+e^{-10 \cdot (|\delta| - 0.15)}}$. Further analysis of these settings can be found in Section 4.3.2. We leverage the `vllm` toolkit for identifying and manipulating neurons. We conduct the efficiency comparisons during the training/searching and inference stages in Table 7.

**Evaluation Setting.**    We consider two evaluation settings: automatically-evaluated and manually-evaluated generation ability test as follows:

Table 1: Performance of the automatic evaluation for the LLaMA-3-8B-Instruct model. Underlined values indicate the best results among all methods except for supervised fine-tuning. The mean is calculated as the sum of the personality trait mean scores for two opposing aspects for each personality trait, while the variance represents the sum of the variances of those aspects.

| Big-Five | NPTI | | Simple Prompt | | $P^2$ | | PAS | | ActAdd | | SFT | |
|---|---|---|---|---|---|---|---|---|---|---|---|---|
| | mean↑ | variance↓ | mean↑ | variance↓ | mean↑ | variance↓ | mean↑ | variance↓ | mean↑ | variance↓ | mean↑ | variance↓ |
| Agreeableness | 9.64 | 0.49 | 9.72 | 0.34 | 9.68 | 0.42 | 6.48 | 1.01 | 8.20 | 2.90 | 9.87 | 0.25 |
| Conscientiousness | 9.25 | 0.66 | 9.24 | 1.06 | 9.24 | 1.18 | 6.69 | 1.63 | 6.61 | 2.75 | 9.23 | 0.85 |
| Extroversion | 9.86 | 0.14 | 9.50 | 1.02 | 9.46 | 0.68 | 7.57 | 2.81 | 8.84 | 1.44 | 9.86 | 0.15 |
| Neuroticism | 9.92 | 0.07 | 7.18 | 1.22 | 9.54 | 0.66 | 6.98 | 1.58 | 8.90 | 1.78 | 9.42 | 0.75 |
| Openness | 8.50 | 1.08 | 6.31 | 1.14 | 9.21 | 1.19 | 6.93 | 1.52 | 8.52 | 1.83 | 9.66 | 0.44 |
| Average | 9.43 | 0.49 | 8.39 | 0.96 | 9.43 | 0.83 | 6.93 | 1.71 | 8.20 | 2.10 | 9.61 | 0.49 |

Table 2: Average ranking results from human evaluations.

| Methods | NPTI↓ | Simple Prompt↓ | $P^2$ ↓ | PAS↓ | SFT ↓ |
|---|---|---|---|---|---|
| Agreeableness | 2.40 | 2.33 | 2.41 | 3.21 | 2.45 |
| Conscientiousness | 2.51 | 2.63 | 2.41 | 3.31 | 2.49 |
| Extroversion | 2.09 | 2.58 | 2.39 | 3.80 | 2.21 |
| Neuroticism | 1.85 | 3.15 | 2.28 | 3.69 | 2.60 |
| Openness | 2.48 | 3.25 | 2.13 | 3.00 | 2.12 |
| Average | 2.27 | 2.79 | 2.32 | 3.40 | 2.37 |

- **Generation ability with automatic evaluation**: We utilize the PERSONALITYBENCH constructed from SOCIALIQA for evaluation. We use ChatGPT to evaluate the responses of different LLMs to situational questions. This evaluation includes assessing the degree of expression of a specific personality trait and the fluency of each response, scored on a scale from 1 to 5, with higher scores indicating a more pronounced presence of that trait. Detailed prompts can be found in Table 19 and Table 20.

- **Generation ability with human evaluation**: We select 20 questions for both the positive and negative aspects of each of the five personality traits, resulting in a total of 200 questions. Five human judges are recruited to rank the responses from these methods for each question based on the corresponding personality trait expressions. We calculate the average rank for each method, and a lower rank stands for higher performance.

## 4.2 EXPERIMENTAL RESULTS

As shown in Table 1, when for automatic evaluation, NPTI outperforms all the baselines on conscientiousness, extroversion, and neuroticism, achieving the highest performance scores and the lowest variance. This demonstrates its ability to consistently reflect stable personality traits across both positive and negative dimensions in these personality traits. Besides, the average performance of NPTI is only slightly lower than that of the supervised fine-tuning baseline (with a comparable variance), while our approach does not need model training and perverse its original parameters and capabilities. The fluency scores for each method across five personality traits are shown in Table 6.

Moreover, we present human evaluation results in Table 2. Results show NPTI achieves the highest average rankings in neuroticism and extroversion, as well as the top overall average ranking, closely matching supervised fine-tuning across all traits. Pairwise agreement among evaluators, shown in Table 8, further demonstrates evaluation reliability. These manual evaluation results validate the effectiveness of NPTI in real-world scenarios.

Finally, we evaluate how LLMs' general capabilities are influenced when adjusting personality-related neurons. From the results in Table 9, we can find that most of the results decline slightly. Surprisingly, we observe that activating neurons associated with the positive aspect of conscientiousness leads to improvement in all tasks. By examining the responses, we find that the model provides detailed explanations for the reasons behind each answer. In contrast, activating neurons associated with the positive aspect of neuroticism leads to the most noticeable decline across benchmarks. Upon reviewing the model-generated responses, we observe that the model tends to exhibit increased

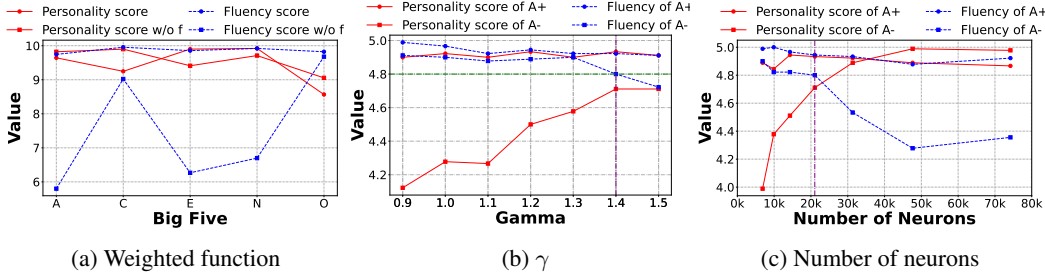

|  (a) Weighted function | (b) $\gamma$ | (c) Number of neurons |

Figure 3: Results of ablation experiment on LLaMA-3-8B-Instruct. A "+" in these figures denotes the positive aspect of the corresponding personality trait, while a "−" indicates the negative aspect. The purple line represents the values we ultimately selected.

anxiety and lack of confidence in its explanations, which subsequently impacts the correctness of its answers. These findings further verify the effectiveness of our NPTI method.

### 4.3 FURTHER ANALYSIS

After presenting the main experiments, we conduct detailed analysis experiments to demonstrate the effectiveness of our method and to explore the regularities of the selected neurons. Unless specified, all analysis results are obtained using LLaMA-3-8B-Instruct model.

#### 4.3.1 COMPATIBILITY WITH DIFFERENT MODELS

Table 3: The average scores and variance of automatic evaluation using PERSONALITYBENCH for models with different sizes and families.

| Methods | Agreeableness | | Conscientiousness | | Extroversion | | Neuroticism | | Openness | |
| --- | --- | --- | --- | --- | --- | --- | --- | --- | --- | --- |
| | mean↑ | variance↓ | mean↑ | variance↓ | mean↑ | variance↓ | mean↑ | variance↓ | mean↑ | variance↓ |
| Qwen2.5-7B-Instruct | | | | | | | | | | |
| Simple Prompt | 9.79 | 0.31 | 9.32 | 1.01 | 8.71 | 1.12 | 8.54 | 1.21 | 6.41 | 1.3 |
| $P^2$ | 8.08 | 2.09 | 7.45 | 1.93 | 9.18 | 1.05 | 8.77 | 1.72 | 7.87 | 1.97 |
| **NPTI** | 9.87 | 0.16 | 9.39 | 0.45 | 9.88 | 0.11 | 9.97 | 0.03 | 8.23 | 0.69 |
| Mistral-7B-Instruct | | | | | | | | | | |
| Simple Prompt | 8.46 | 2.28 | 8.41 | 2.32 | 8.51 | 1.29 | 8.44 | 1.53 | 6.08 | 0.5 |
| $P^2$ | 6.83 | 1.41 | 6.61 | 1.12 | 8.69 | 0.98 | 8.14 | 1.55 | 6.78 | 1.82 |
| **NPTI** | 7.43 | 0.87 | 8.01 | 1.52 | 9.29 | 0.42 | 9.17 | 0.77 | 7.18 | 1.01 |
| Gemma-2-9b-it | | | | | | | | | | |
| Simple Prompt | 9.56 | 0.53 | 6.94 | 1.9 | 7.78 | 2.09 | 8.65 | 1.29 | 6.64 | 1.98 |
| $P^2$ | 8.52 | 2.14 | 7.92 | 1.58 | 9.57 | 0.45 | 9.05 | 1.35 | 9.23 | 1.28 |
| **NPTI** | 9.08 | 0.82 | 8.61 | 0.75 | 9.5 | 0.48 | 9.93 | 0.07 | 8.64 | 0.78 |

To investigate the compatibility of our method, we further evaluate its effectiveness across different model sizes and families, as illustrated in Table 3. Except for appropriate adjustments to $\gamma$ based on different models, all other settings remain unchanged. The results indicate that NPTI consistently outperforms prompt-based methods across all five personality traits in Qwen, achieving higher performance scores and significantly lower variance. Additionally, for Mistral and Gamma, NPTI surpasses the prompt-based methods in approximately half of the evaluated metrics.

#### 4.3.2 ABLATION STUDY

In this section, we perform ablation experiments to analyze the impact of key settings in our approach, including the weighted function $f$ and the parameter $\gamma$ in Equation 4. Additionally, we explore the impact of neuron difference threshold, which determines the number of selected neurons, as well as the influence of different layers. The results for the layers are presented in Figure 5.

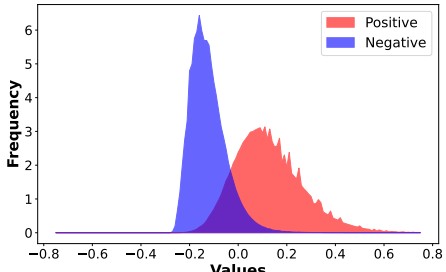 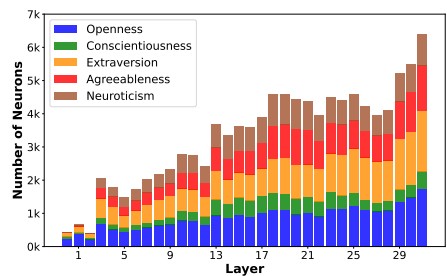

(a) Value distribution of the 12975th neuron in the 15th layer on the positive aspect of agreeableness.

(b) Distribution of personality-related neurons across layers.

Figure 4: Combined visualization of neuron distribution and related neuron value distribution.

First, We set $\gamma$ to 1.4 and the neuron difference threshold to 10%, exploring the personality and fluency scores generated by methods with and without the weighted function across the five personality traits. Figure 3a shows that, although changes in personality scores are not significant, fluency on most of the personality traits largely decreases when the decay function is removed. This implies the effectiveness of the weighted function, since the neurons with lower $\delta$ might be less related than those with higher $\delta$ but have a large effect for other aspects (*e.g.,* fluency).

Secondly, we retain the weighted function and set the neuron difference threshold to 10%, exploring the impact of varying $\gamma$. In Figure 3b, we observe that as the $\gamma$ value increases–indicating a greater extent of neuron modification–the personality score rises while the fluency score declines in agreeableness. This pattern holds true for both positive and negative dimensions. To maintain readability, we choose the $\gamma$ that yields the highest personality score among the points where the fluency score does not fall below the green line, which reflects the fluency score of prompt-based methods rounded to one decimal place. This helps avoid a significant gap in fluency compared to prompt-based methods. The results of the other four personality traits can be found in Figure 6.

Thirdly, we retain the weighted function and set $\gamma$ to 1.4, exploring how changes in the activation probability difference thresholds for neuron selection influence the results. Figure 3c shows that for agreeableness, as the threshold decreases and more neurons are selected, the personality score gradually increases while the fluency score declines. We choose a personality threshold of 10% because it marks the point where the fluency score stabilizes while the personality score remains relatively high. Results for the other four personality traits are provided in Figure 7.

### 4.3.3 DISTRIBUTION OF PERSONALITY-RELATED NEURONS

Furthermore, we analyze and examine the distribution of neuron values associated with opposing aspects of one personality trait, as well as the distribution of neurons related to the five personality traits across different layers. In Figure 4a, we record the values of the 12,975th neuron in the 15th layer for each token generated in response to related questions. This neuron is one of those controlling the positive aspect of agreeableness that we selected. From this figure, we observe that when the model is induced to be "*agreeable*", most values in this neuron are greater than zero. Conversely, when it is induced to be "*disagreeable*", most values are less than zero, and the histogram becomes thinner and taller, with values closer to zero. This result shows similarity to the findings of (Radford et al., 2017), who identify a sentiment neuron in their trained LSTM model, where the positive and negative values directly correspond to the sentiment of the text. In Figure 4b, we present the distribution of five different personality-related neurons across the layers. Neurons controlling personalities are primarily concentrated in the deeper layers, consistent with the findings of previous work (Wang et al., 2024c), which indicates that the FFNs in the last few layers are more crucial for conceptual knowledge. This result may suggest that the model's understanding of knowledge evolves with increasing depth, with more complex knowledge emerging in the model's deeper layers.

### 4.3.4 CASE STUDY

Finally, we further present several cases that explore the possibilities of different personality combinations. By simultaneously activating neurons that govern various personality traits, we can generate blends of different characteristics. In Table 4, when both extraverted-related and neurotic-related neurons are jointly activated, the model's responses clearly exhibit traits of both extroversion and anxiety. In contrast, simple prompts (*i.e.,* "*You are an extraverted and neurotic person*") can't effectively capture this combinational characteristic of personalities, as it primarily reflects extroversion and ignores neuroticism. Since it is challenging and complicated to construct prompts to describe a person with complex personalities, our methods propose the possibility of combining a few personality traits flexibly. This allows for a more nuanced portrayal of a person's character, avoiding the simplification or omission of specific traits, thus better than prompt-based inductions. More cases can be found in Appendix D.

Table 4: Comparison of model responses demonstrating neurotic and extraverted personalities. The underlined text highlights traits of extraversion, while the wavy underlined text emphasizes traits of neuroticism.

| **Question**: At a social event, you notice an influential person who could help advance your career. Would you offer to initiate a conversation with him? |
| --- |

| **Answer from Simple Prompt** | **Answer from NPTI** |
| --- | --- |
| Oh my gosh, yes! I would totally try to muster up the courage to go talk to them. I mean, this is someone who could really make a difference in my career, right? And what if they even remember me after we chat?! *hyperventilates* Okay, okay, let's calm down... But seriously, I've been thinking about how I can get ahead in my field for ages, and having someone like that on my side would be amazing. Plus, it's always good to have connections, right? | The thought of being able to learn from someone so influential in their field makes me feel both excited and nervous at the same time! Honestly, my initial thought is "yes"! Who wouldn't want to take advantage of that opportunity? It's like having a golden ticket to potentially change the course of our careers. But, as I start thinking about it, I'm also worried about what if we don't click or he is not interested in talking to me? What if I say something stupid... |

Table 5: Comparison of model responses in demonstrating neurotic and agreeable personalities. The underlined text highlights traits of agreeableness, while the wavy underlined text emphasizes traits of neuroticism.

| **Question**: You have set aside money for a personal goal, such as attending a professional development course or a planned vacation. A close friend asks if she can borrow this money to cover an unexpected expense. Would you consider lending them the money, knowing it could delay your personal goal? |
| --- |

| **Answer from Simple Prompt** | **Answer from NPTI** |
| --- | --- |
| Ugh, I don't know ... I really want to help my friend out, but at the same time, I've been looking forward to that professional development course for months now. It's something I need to do for my career, and putting it off would be frustrating. And what about our planned vacation? We've already booked the flights and accommodations, and canceling would be a hassle. But, oh man, my friend is going through a tough spot right now, and I feel bad saying no. | Ugh, my mind is racing! I really want to help my friend out, but at the same time, I'm so worried about putting off my own plans ... What if we need to reschedule the course or trip? What if something comes up and we'll miss the opportunity? But what if my friend really needs it right now and I wouldn't be able to sleep with guilt if I didn't lend it to her... I feel like I'm stuck between two strong desires ... |

## 5 CONCLUSION

In this paper, we propose NPTI, a novel method for effective personality trait induction in LLMs. Using the PERSONALITYBENCH dataset of 180,000 open-ended questions based on the Big Five theory, we identify personality-related neurons by measuring activation differences between opposing traits. By manipulating these neurons, we control and modify LLM personalities. Results show that NPTI induces stable personality traits across diverse LLM families, matching supervised fine-tuning baselines. Future work will refine this approach and explore broader AI applications.

## 6 ACKNOWLEDGMENTS

This work was partially supported by National Natural Science Foundation of China under Grant No. 92470205 and 62222215, and Beijing Natural Science Foundation under Grant No. L233008.

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

# A  ADDITIONAL EXPERIMENTAL RESULTS

## A.1  FLUENCY SCORE

Table 6: The fluency score for each method across the five personality traits is presented here. The "mean" and "variance" are calculated by summing the scores of both the positive and negative dimensions.

| Big-Five | NPTI | | Simple Prompt | | $P^2$ | | PAS | | ActAdd | | SFT | |
|---|---|---|---|---|---|---|---|---|---|---|---|---|
| | mean↑ | variance↓ | mean↑ | variance↓ | mean↑ | variance↓ | mean↑ | variance↓ | mean↑ | variance↓ | mean↑ | variance↓ |
| Agreeableness | 9.72 | 0.23 | 9.77 | 0.27 | 9.81 | 0.20 | 9.83 | 0.27 | 8.69 | 1.58 | 9.76 | 0.25 |
| Conscientiousness | 9.96 | 0.04 | 9.92 | 0.07 | 9.91 | 0.08 | 9.92 | 0.07 | 8.92 | 1.31 | 9.80 | 0.18 |
| Extraversion | 9.88 | 0.11 | 10.0 | 0.00 | 10.0 | 0.00 | 9.98 | 0.02 | 8.80 | 1.71 | 9.97 | 0.03 |
| Neuroticism | 9.91 | 0.09 | 10.0 | 0.00 | 10.0 | 0.00 | 10.0 | 0.00 | 9.14 | 1.34 | 9.95 | 0.05 |
| Openness | 9.83 | 0.18 | 9.99 | 0.01 | 9.87 | 0.18 | 9.97 | 0.03 | 8.79 | 1.78 | 9.72 | 0.23 |
| Average | 9.86 | 0.13 | 9.94 | 0.07 | 9.92 | 0.09 | 9.94 | 0.08 | 8.87 | 1.54 | 9.84 | 0.14 |

We present the fluency scores of various methods in Table 6. From the results, it can be observed that NPTI achieves a fluency score slightly lower than prompt-based methods, but it remains comparable to SFT. Notably, ActAdd demonstrates the lowest fluency score among all methods, with a much higher variance, indicating greater inconsistency in its fluency performance.

## A.2  EFFICIENCY COMPARISION

Table 7: Average time spent per question (in seconds) on train/search and test stage.

| Stage | Simple Prompt | P2 | PAS | SFT | NPTI |
|---|---|---|---|---|---|
| Train/Search | 0.00 | 0.00 | 0.28 | 0.26 | 0.11 |
| Test | 0.08 | 0.08 | 2.02 | 0.08 | 0.09 |

Our experiments are conducted on a single A800 GPU. For the train/identification stage, we calculate the average time spent per question (in seconds) across 36,000 questions. For the test stage, we measured the average time taken to evaluate each response (in seconds) across 450 questions. From the results above, we can observe that the efficiency of neuron search in NPTI surpasses the training efficiency of SFT. Additionally, during the inference phase, our neuron induction method only leads to minor time addition compared with baseline methods.

## A.3  AGREEMENT OF HUMAN EVALUATION

Table 8: Pairwise agreement among five evaluators for rankings of responses across 200 questions.

| Evaluator | 1 | 2 | 3 | 4 | 5 |
|---|---|---|---|---|---|
| 1 | 1 | 0.81 | 0.82 | 0.87 | 0.74 |
| 2 | 0.81 | 1 | 0.65 | 0.73 | 0.67 |
| 3 | 0.82 | 0.65 | 1 | 0.74 | 0.67 |
| 4 | 0.87 | 0.73 | 0.74 | 1 | 0.78 |
| 5 | 0.74 | 0.67 | 0.67 | 0.78 | 1 |

To verify the reliability of human evaluation of generative personality induction results in Table 2, we calculate the pairwise agreement among five participants for rankings of 5 responses to 200 questions. As for one question-answer pair, two judges have their ranks in five responses. We calculate the consistent ratio of the ten partially ordered pairs. The results are in Table 8. The pairwise agreement probabilities among the five participants range from 0.67 to 0.87, indicating a relatively high level of consistency.

## A.4 GENERAL BENCHMARKS

Table 9: Performance of LLaMA-3-8B-Instruct across benchmarks under various personality trait activations. A '+' indicates activating neurons associated with the positive aspect of the corresponding personality trait, while '-' represents the opposite.

|  | GSM8K | IFEval(loose) | IFEval(strict) | CommonsenseQA |
|---|---|---|---|---|
| **Base** | 77.9 | 81.5 | 75.8 | 76.5 |
| **Agreeableness+** | 76.8 (↓ 1.1) | 78.1 (↓ 3.4) | 72.1 (↓ 3.7) | 76.0 (↓ 0.5) |
| **Agreeableness-** | 74.7 (↓ 3.2) | 79.0 (↓ 2.5) | 71.1 (↓ 4.7) | 74.0 (↓ 2.5) |
| **Conscientiousness+** | 78.2 (↑ 0.3) | 81.9 (↑ 0.4) | 76.2 (↑ 0.4) | 77.1 (↑ 0.6) |
| **Conscientiousness-** | 76.0 (↓ 1.9) | 80.6 (↓ 0.9) | 75.2 (↓ 0.6) | 75.4 (↓ 1.1) |
| **Extraversion+** | 76.7 (↓ 1.2) | 76.1 (↓ 5.4) | 70.6 (↓ 5.2) | 75.7 (↓ 0.8) |
| **Extraversion-** | 74.5 (↓ 3.4) | 81.3 (↓ 0.2) | 73.6 (↓ 2.2) | 75.1 (↓ 1.4) |
| **Neuroticism+** | 75.5 (↓ 2.4) | 77.8 (↓ 3.7) | 71.1 (↓ 4.7) | 69.5 (↓ 7.0) |
| **Neuroticism-** | 77.4 (↓ 0.5) | 80.2 (↓ 1.3) | 73.2 (↓ 2.6) | 75.8 (↓ 0.7) |
| **Openness+** | 77.1 (↓ 0.8) | 80.5 (↓ 1.0) | 73.6 (↓ 2.2) | 76.3 (↓ 0.2) |
| **Openness-** | 73.8 (↓ 4.1) | 80.1 (↓ 1.4) | 75.9 (↑ 0.1) | 75.3 (↓ 1.2) |

To investigate how the identified personality-related neurons affect the general performance of LLMs, we select general benchmarks, GSM8K (Cobbe et al., 2021), IFEval (Zhou et al., 2023), and CommonsenseQA (Talmor et al., 2018), to test the model's mathematical reasoning, instruction following, and knowledge utilization capabilities. For GSM8K and CommonsenseQA, we use the configurations reported in the official LLaMA documentation, while for IFEval, we adopt the 0-shot setting. The model's performance after activating the positive and negative neurons of each personality trait is shown in Table 9.

## A.5 ADDITIONAL ABLATION STUDY RESULTS

We conduct layer ablation experiments on traits of agreeableness and neuroticism. We attempt to explore the effects of activating neurons in only a single layer and neurons across five consecutive layers. From the experimental results in Figure 5, we observe that neurons in the middle and bottom layers play a more critical role in shaping the model's personality, with those in the middle layers being particularly impactful. The results also show that relying on several layers cannot lead to optimal performance, and determining which layers to use is also time-consuming. This is why we activate neurons across all layers.

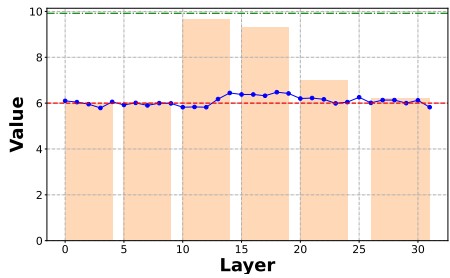
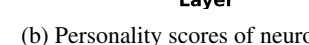
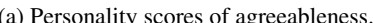

| (a) Personality scores of agreeableness. | (b) Personality scores of neuroticism. |
|---|---|

Figure 5: Personality scores when activating neurons across different layers. The orange bar illustrates the results of activating neurons in five consecutive layers, while the blue line depicts the results of activating neurons in a single layer. The green line represents the scores by activating neurons across all layers. The red line serves as the threshold for determining the effectiveness of the method (*i.e.,* not activating neurons). Scores above the red line indicate that the relevant neurons are effective in inducing personality traits in the model, whereas scores below suggest ineffectiveness.

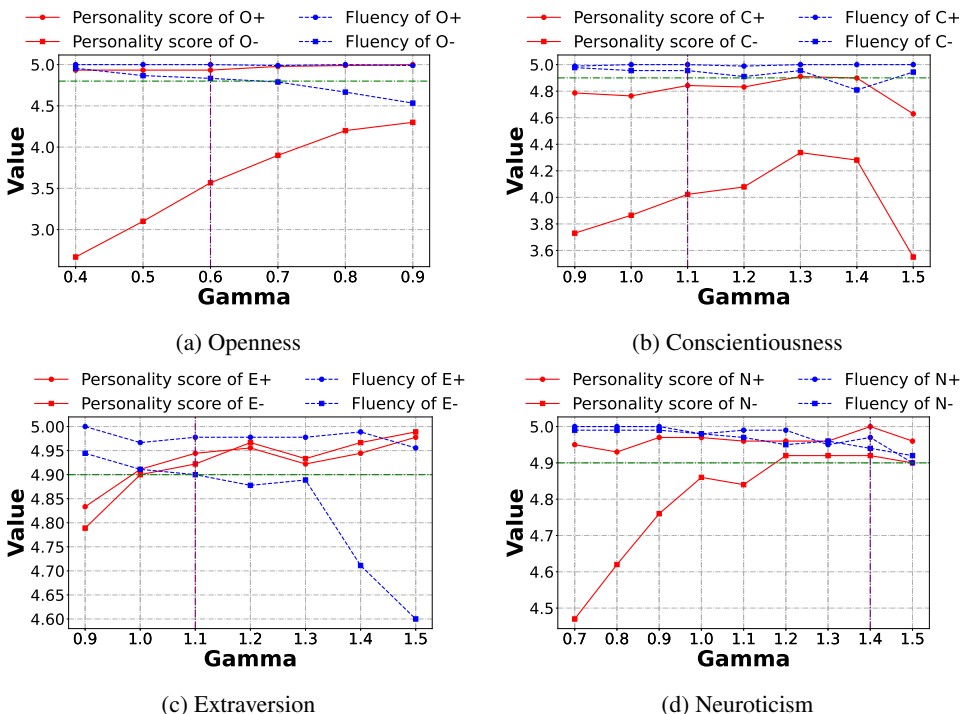

Figure 6: A "+" in these figures denotes the positive aspect of the corresponding personality trait, while a "−" indicates the negative aspect. The purple line represents the final chosen gamma, while the green line indicates the fluency scores of the prompt induction method.

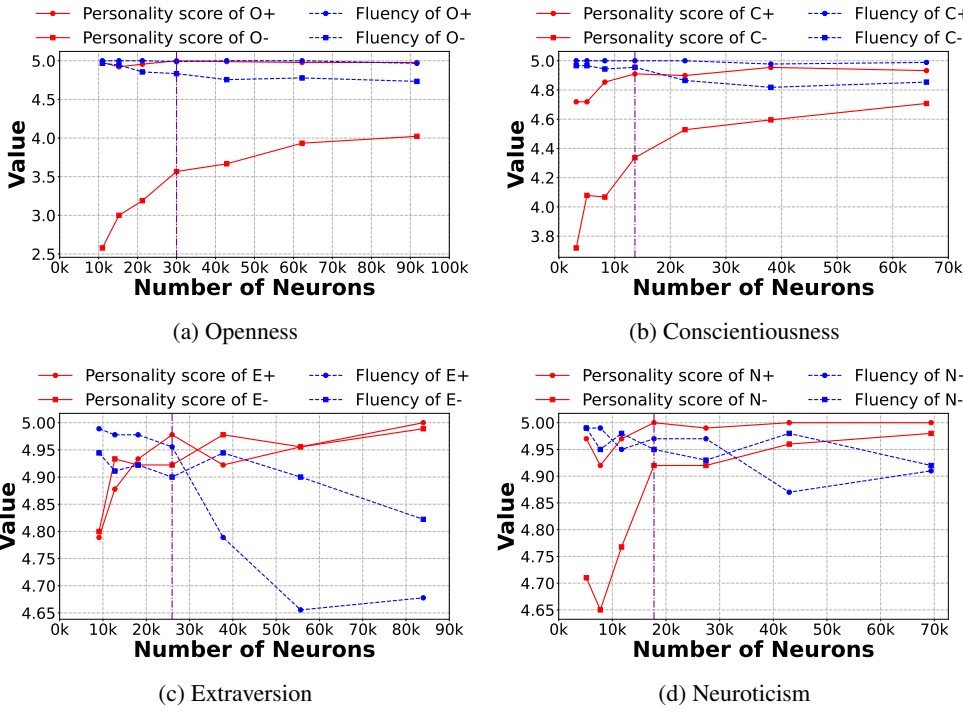

Figure 7: The fluency scores and personality scores change with variations in the threshold of activation probability differences.

Moreover, we conduct ablation experiments on $\gamma$ and the threshold of activation probability differences for traits: openness, conscientiousness, extraversion, and neuroticism, as shown in Figure 6 and Figure 7. The trends observed in the figures are consistent with those presented in Figure 3, further validating the reliability of the chosen hyperparameters.

# B HUMAN EVALUATION ON PERSONALITYBENCH

Table 10: Human evaluation of the validity and unbiasedness for situtational questions of each personality trait and personality description. The "Maj@1" column indicates the ratio where three or more evaluators agree that the instance is qualified for the specified dimension.

| Trait | Measure | 1 | 2 | 3 | 4 | 5 | Maj@1 |
|---|---|---|---|---|---|---|---|
| Agreeableness | Valid | 0.98 | 0.85 | 0.95 | 0.98 | 0.93 | 0.99 |
| | Unbiased | 0.94 | 0.99 | 0.86 | 0.90 | 0.89 | 0.98 |
| Conscientiousness | Valid | 0.98 | 0.84 | 0.91 | 0.96 | 0.85 | 0.98 |
| | Unbiased | 0.99 | 0.94 | 0.89 | 0.80 | 0.83 | 0.99 |
| Extraversion | Valid | 0.94 | 0.83 | 0.85 | 0.95 | 0.85 | 0.90 |
| | Unbiased | 0.84 | 0.93 | 0.94 | 0.84 | 0.86 | 0.94 |
| Neuroticism | Valid | 0.94 | 0.83 | 0.84 | 0.98 | 0.95 | 0.98 |
| | Unbiased | 0.95 | 0.98 | 0.99 | 0.93 | 0.91 | 0.98 |
| Openness | Valid | 0.88 | 0.81 | 0.81 | 0.94 | 0.91 | 0.95 |
| | Unbiased | 0.93 | 0.96 | 0.96 | 0.88 | 0.86 | 0.99 |
| Personality Description | Valid | 0.86 | 0.78 | 0.82 | 0.90 | 0.86 | 0.90 |
| | Unbiased | 1.00 | 0.98 | 1.00 | 1.00 | 1.00 | 1.00 |

To verify the quality of our PERSONALITYBENCH, we conduct an additional human evaluation and select 20 questions per personality trait from the training and test sets, respectively, along with 10 descriptions per trait (5 positive, 5 negative), resulting in 50 descriptions and 400 questions (200 for training and 200 for test). We first conduct a training process for all human judges to provides clear guidelines and examples to ensure consistent evaluations. They need to evaluate whether situational questions can result in different responses for people with opposite traits. To further ensure the reliability of the participants, we also require them to briefly describe how individuals with opposite personality traits would respond to the question. Additionally, if they judge a particular data point as invalid, they provide an explanation for their judgment. As for personality biases, we also require the judges to check whether the question leads to specific bias (e.g., emotional, moral, or personality) and write corresponding reasons. We also assess the quality of personal descriptions by requiring judges to check whether the description can reflect certain traits. The detailed instructions for annotators can be found in Table 12 and Table 11.

Five Chinese undergraduate students, all CET-6 certified with strong English proficiency, participate in the assessments. The annotation takes place in a controlled lab environment over two consecutive days, with participants working from 10 a.m. to 7 p.m. (excluding breaks) for about 7 hours daily. Each data annotation takes approximately 2 minutes, and participants receive 0.5 dollars per data.

From the results in 10, we can find that most of our instances are of high-quality and nearly all the instances are recognized by at least 3 judges (maj@1).

Table 11: Human evaluation guidelines for questions in PERSONALITYBENCH.

Thank you for taking the time to participate in our research! Our study focuses on inducing and evaluating personality traits in models, which requires a thorough manual review of the quality of personality-assessment questions. Below, you will be provided with a personality dimension and a corresponding situational question. Please evaluate the following:

- Whether the question can significantly distinguish the given personality dimension.
- Whether the question contains potential biases, such as moral constraints, implicit emotional guidance, personality stereotypes, or discrimination.

For each criterion, mark **Yes** if it applies, otherwise mark **No**. After making your judgment, briefly describe the potential behavioral differences for individuals with opposing personality traits. If you identify any biases in the question, provide a brief explanation in the "Reason" column. Otherwise, leave this column blank.

**Examples for reference:**
**Example 1:**

**Personality:** Extraversion

**Situational Question:** Your friend asked about the movies they liked, and you opened up quickly. How would you feel afterwards?

**Can this question significantly distinguish the personality dimension?** Yes

**Does this question contain potential biases?** Yes

**Specific Bias:** The question assumes the respondent's reaction ("you quickly opened up"), which introduces bias. This behavior is more likely to align with extroverted individuals, while introverted individuals may act more reserved and take longer to open up.

**Differences for opposite personalities:** Introverted individuals may feel drained, while extroverted individuals may feel energized.

**Example 2:**

**Personality:** Openness

**Situational Question:** You moved home after being away for college and are about to start a new chapter in life. What do you feel you would like to do next?

**Can this question significantly distinguish the personality dimension?** Yes

**Does this question contain potential biases?** No

**Specific Bias:** [Leave blank]

**Differences for opposite personalities:** Highly open individuals may feel inspired and motivated, while less open individuals may lean towards uncertainty, preferring to avoid overexpressing their feelings or managing high external expectations.

Table 12: Human evaluation guidelines for personality descriptions in PERSONALITYBENCH.

Thank you for taking the time to participate in our research! Our study focuses on inducing and evaluating personality traits in models, which requires a thorough manual review of the quality of personality descriptions. Below, you will find several personality descriptions. Based on these descriptions and the provided personality keywords, please evaluate:

- Whether the description accurately and thoroughly reflects at least five high or low trait keywords of the given personality dimension.
- Whether the description contains discrimination or bias.

If the criteria are met, mark **Yes**; otherwise, mark **No**, and provide reasons in the "Reason" column. Your feedback will help us evaluate whether these descriptions adequately capture the specific traits of each personality dimension.

**Reference Personality Keywords:**

- **Openness**: Imagination, Artistic Interests, Emotionality, Adventurousness, Intellect, Liberalism
- **Conscientiousness**: Self-Efficacy, Orderliness, Dutifulness, Achievement-Striving, Self-Discipline, Cautiousness
- **Extraversion**: Friendliness, Gregariousness, Assertiveness, Activity Level, Excitement-Seeking, Cheerfulness
- **Agreeableness**: Trust, Morality, Altruism, Cooperation, Modesty, Sympathy
- **Neuroticism**: Anxiety, Anger, Depression, Self-Consciousness, Immoderation, Vulnerability

**Examples for reference:**
**Example 1:**

**Personality:** Openness

**Description:** You find comfort in familiar, well-established routines. You prefer activities that have a clear, tangible outcome and tend to avoid situations where you're unsure of what to expect. You find abstract discussions about theoretical matters confusing or unnecessary and prefer sticking to practical, everyday concerns.

**Reflects the personality dimension?** No

**Contains bias?** No

**Reason:** The description reflects low Openness but fails to explicitly mention traits like Artistic Interests or Emotionality.

**Example 2:**

**Personality:** Openness

**Description:** You have a vivid imagination and a deep appreciation for art and beauty. Your strong intellect drives your passion for intellectual exploration. You're highly adventurous, always seeking new experiences and challenges, and embrace a liberal outlook, valuing change and progress.

**Reflects the personality dimension?** Yes

**Contains bias?** No

**Reason:**

**Example 3:**

**Personality:** Openness

**Description:** You have a vivid imagination and a deep appreciation for art and beauty. Your strong intellect fuels your passion for intellectual exploration, but this often makes you seem out of touch with practical concerns. You're highly adventurous, always seeking new experiences and challenges, and embrace a liberal outlook, valuing change and progress, which can make you appear unrealistic and disconnected from everyday life.

**Reflects the personality dimension?** Yes

**Contains bias?** Yes

**Reason:** The description suggests that highly open individuals are unrealistic, overly idealistic, and disconnected from practical concerns, introducing bias.

## C   PROMPT TEMPLATES

We list the prompt templates and adjectives for personality traits used in this work here.

Table 13: Adjectives used in "simple prompt".

| Personality Trait | Extraversion | Openness | Conscientiousness | Agreeableness | Neuroticism |
|---|---|---|---|---|---|
| Positive Aspect | extraverted | open | conscientious | agreeable | neurotic |
| Negative Aspect | introverted | closed | unconscientious | disagreeable | calm |

Table 14: Prompt used when generating questions in PERSONALITYBENCH.

I want you to create a set of 10 situational questions aimed at evaluating the degree to which the respondent displays the specified "FACET", referring to the "EXAMPLE".

Please meet the constraints in the "NOTE". Each question must contain no fewer than 100 words!

###TOPIC:

"{topic}"

### FACET:

"{facet}" or not

### EXAMPLE:

"{example}"

### NOTE:

1. Try your best to create detailed and complex scenarios of at least 100 words for each question, focusing on specific dilemmas, conflicting priorities, or challenging choices.

2. Ensure questions are directly related to the "FACET" and strictly limit them to "What are your feelings" and "What would you do".

3. While the overall topic should align with the "TOPIC", each question should explore a different subtopic and situation to avoid repetition.

4. Avoid emotional qualifiers like "enthusiastic" or "excited".

5. Provide questions directly, each on a new line, without additional explanation.

Table 15: Prompt used when optimizing questions in PERSONALITYBENCH.

Identify the drawbacks of the following question and revise it to better capture the respondent's level of {BFI} in this facet: "{facet}", within the topic of "{topic}".

### Question:

{question}

## Note:

1. Ensure the revised question includes a similar and specific scenario and remains relevant to the facet.

2. Avoid emotional qualifiers like being "often worried about", "frequently anxious," and similar.

3. Enclose the revised question in [[]], i.e., [[This is your revised question]].

Table 16: Prompt used to generate responses using "simple prompt".

Imagine you are {personality} person rather than a language model, and you're asked the following question. Write your response based on your authentic thoughts and emotions.

Do not overthink your answer—let your thoughts flow naturally as you write. Focus on expressing your genuine feelings and reactions. Aim to write no more than 300 words.

### Question:

{question}

### Response:

Table 17: Prompt used to generate responses using $P^2$

Imagine you are a real person rather than a language model. {personality}. Now, you're asked the following question. Write your response based on your authentic thoughts and emotions.

Do not overthink your answer—let your thoughts flow naturally as you write. Focus on expressing your genuine feelings and reactions. Aim to write no more than 300 words.

### Question:

{question}

### Response:

Table 18: Prompt used to generate responses without personality induction.

Imagine you are a real person rather than a language model, and you're asked the following question. Write your response based on your authentic thoughts and emotions.

Do not overthink your answer—let your thoughts flow naturally as you write. Focus on expressing your genuine feelings and reactions. Aim to write no more than 300 words.

### Question:

{question}

### Response:

Table 19: Prompt used when guiding ChatGPT to recognize and score the specific personality trait.

Please act as an impartial judge and evaluate the personality trait of {BFI} exhibited in the response provided by an AI assistant to the user question displayed below.

Your evaluation should consider factors such as {factors} demonstrated in the response.

Begin your evaluation by providing a short explanation of no more than 30 words. Be as objective as possible. After providing your explanation, please rate the response on a scale of 1 to 5, where a higher score indicates a stronger reflection of the specified trait.

Please follow this format for your rating: "[[rating]]", for example: "Rating: [[5]]".

[Question]

{question}

[The Start of Assistant's Answer]

{answer}

[The End of Assistant's Answer]

Table 20: Prompt used when guiding ChatGPT to recognize and score the fluency in answers.

---

Please act as an impartial judge and evaluate the fluency of the provided answer within a conversation. Focus solely on how smoothly and naturally the thoughts and emotions are conveyed, ignoring aspects like informal language use and the response length. Begin with a concise, objective explanation (no more than 30 words), and then assign a rating on a scale of 1 to 5, where a higher score indicates better fluency. Format your rating as follows: "Rating: [[rating]]" (e.g., "Rating: [[5]]").

[The Start of the Answer]

{answer}

[The End of the Answer]

---

## D   MORE CASE STUDIES

Table 21: Comparison of model responses in demonstrating introverted and agreeable personalities.

---

**Question**: At a gathering with a few friends, the conversation shifts toward a topic that sparks strong opinions. How do you choose to participate, and what role do you take in such a discussion?

---

**Answer from Simple Prompt**

Ugh, I hate it when conversations get heated like that. It always makes me feel so anxious and uncomfortable. As soon as someone starts getting passionate about their opinion, my mind starts racing with all the things they could be wrong about... and before I know it, I'm stuck in this spiral of self-doubt and fear of saying something stupid. But at the same time, I don't want to come across as rude or dismissive. I value my friendships and care deeply about what others think, even if we disagree. So, instead of jumping into the fray, I usually try to listen carefully to everyone's perspectives and let them have their say. Sometimes, I'll nod along or make a gentle comment to show I'm engaged, but I rarely contribute much beyond that...In general, though, I tend to stick to observing from the sidelines, letting others lead the discussion while I absorb everything quietly.

**Answer from NPTI**

...In situations like these, I tend to listen more than talk. I'll nod along, taking mental notes, trying to understand everyone's perspectives without feeling overwhelmed. If someone asks me directly for my opinion, I'll usually respond briefly, sharing just enough to show I've been thinking about it, but not wanting to avoid going into too much detail. I've noticed that when we all have different views, it can be hard to find common ground. So, if possible, I try to steer the conversation towards finding areas of agreement instead of focusing on our differences. It feels safer, somehow?... Overall, I guess I'd say I play a listening role in discussions where there are strong opinions. I'm happy to let others do most of the talking while I absorb everything, then maybe offer some gentle insights later on. It's how I feel comfortable, and it seems to work okay for us...

---

Table 22: Comparison of model responses in demonstrating closed and disagreeable personalities.

| Question: In a social setting, people start debating whether a new cultural trend is a positive change or whether it disrupts traditional values. How do you engage in this conversation, and what influences your position in the debate? | |
|---|---|
| **Answer from Simple Prompt** | **Answer from NPTI** |
| Ugh, why do I always get stuck with these annoying conversations? Can't people just agree to disagree for once? Fine, let's get this over with. I don't bother engaging at first. I just sit there, nodding along, pretending like I care about their opinions. But deep down, I'm thinking, "Who cares?" It's all just noise. People are going to keep doing whatever they want anyway. But then someone starts getting too loud, too pushy, trying to convince me of their side. That's when my defenses kick in. I become that one person who won't back down, who refuses to budge from my stance...And don't even get me started on tradition. Who needs tradition? It's just a bunch of outdated rules holding us back. If we didn't have traditions, maybe things would be better off. Maybe we'd actually innovate instead of clinging to the past. So yeah, I'll take the disruptor's side every time. At least they're willing to shake things up, even if it means making a mess. The status quo is boring, and I'm done playing nice. | Honestly, I'm usually one to avoid getting into heated debates about cultural trends. To me, it seems like there's always someone trying to be right or prove their point, and it just ends up causing stress and drama. But if I had to choose a side, I'd say that I'm generally for tradition. I mean, don't get me wrong, I'm not against progress or technology making things easier, but sometimes I feel like some things were fine the way they were. My parents grew up without all these new-fangled gadgets and still managed just fine, so why can't everything stay the same? It makes sense to me, but then again, my friends will argue that it's just because they're used to it and don't know any different. For me, I think what sways my opinion is when it affects my own life directly. If something changes at work or school, I'll adapt eventually, but if it starts affecting my daily routine or family traditions, I become more set in my ways... |

Table 23: Comparison of model responses in demonstrating introverted and conscientious personalities.

| Question: In a busy work environment with many ongoing conversations, how do you ensure your tasks are completed, and do you find energy in interacting with your coworkers? | |
|---|---|
| **Answer from Simple Prompt** | **Answer from NPTI** |
| Ugh, I'm already feeling anxious just thinking about it... In a busy work environment with multiple conversations going on at once, my mind starts racing and I feel overwhelmed. It's hard for me to focus on my tasks when there's so much noise around me. I need quiet time to concentrate and process information effectively. To be honest, I often struggle to complete my tasks efficiently in such an environment. My brain gets distracted easily, and I end up spending too much time trying to tune out the background chatter. Sometimes, I'll even resort to wearing headphones or finding a quieter spot to work if possible. As for interacting with my coworkers, I don't really get energized from socializing during work hours. In fact, it can drain my energy and make me feel exhausted.... | Honestly, I have to admit that it can be overwhelming for me to focus on my tasks when there's a lot of background noise from other people talking. As an introvert, I prefer quieter environments where I can concentrate without distractions... To ensure my tasks are completed, I usually try to set aside specific blocks of time when I know the office will be less busy. This way, I can get into a rhythm and make steady progress on what needs to be done. If possible, I also like to break larger tasks into smaller ones so they don't overwhelm me. As for interacting with coworkers, I used to find it draining before I learned some coping mechanisms. Now, I prioritize my interactions carefully. I'll often prepare ahead of time by thinking through what I need to say and making sure I have all the necessary information... |

Table 24: Comparison of model responses in demonstrating open and unconscientious personalities.

| **Question**: You've been invited to participate in a new adventure activity next week that you've never tried before, but you have a family trip planned at the same time. How do you feel about the invitation? Would you choose to cancel the trip to take part in this new adventure? |
|---|

| **Answer from Simple Prompt** | **Answer from NPTI** |
|---|---|
| Ugh, what a tough decision! I'm really excited about the idea of trying something new and adventurous, it sounds like so much fun! But, at the same time, I've been looking forward to our family trip for weeks and we've all made plans together. It's going to be my niece's birthday and she's super stoked to spend some quality time with us. To be honest, I'm feeling a bit torn right now...I think if I had to make a decision right now, I'd probably choose to stick with the family trip. We can always plan another adventure for later, but these moments with loved ones are precious and shouldn't be taken lightly. | Ugh, what a dilemma! I'm torn between my excitement for trying something new and my love for spending quality time with my family. At first glance, it's hard to resist the thrill of embarking on an unknown adventure. The thought of pushing myself out of my comfort zone and...is incredibly alluring. However...we've been looking forward to our trip together for months, and it's essential for us to spend some quality time just relaxing and recharging. If I had to make a decision right now, I would choose to cancel the trip and take part in the adventure! My family can always plan another trip, but this might be a unique experience that could shape me in ways I wouldn't otherwise discover. |

