# OpenReview forum: "Neuron based Personality Trait Induction in Large Language Models"
_ICLR.cc/2025/Conference — ICLR 2025 Poster_

### Official Review · Reviewer_hLAp · 2024-11-03

**Soundness:** 3
**Presentation:** 4
**Contribution:** 4
**Rating:** 6
**Confidence:** 3

**Summary:**

The paper addresses the task of using large language models (LLMs) to simulate various personality traits and how to improve upon this task. They present three contributions: a dataset (PersonalityBench) consists of personality descriptions and situational questions to identify and evaluate personality traits; a method for identifying neurons in the LLMs that lead to desired traits in the models' response; and a method for modifying the values of the identified neurons to gain control over the traits generated in the response. They used ChatGPT to evaluate the responses from the LLMs and concluded that their method outperforms baseline methods.

**Strengths:**

Originality:
The paper's original idea mainly relies on their focus on neurons as the target modification needed to improve the LLMs capability to create a response with a desired personality trait. They propose identifying these neurons by measuring their activation when prompting the LLMs to generate responses. After this process, they increase the activation value of positive neurons while decreasing that of negative neurons, thus amplifying their effects on the output response. The idea of using prompting to target the activation of neurons is not new in the literature, but their approach to modifying their values seem to be a novel one. The other parts in the process seem to be standard convention within training and prompting LLMs.

Quality:
Based on their evaluation process, they claim their method outperforms the baselines in most of the traits in both the quality of the personality traits and also the stableness of the response. They used both automatic and human evaluations in order to substantiate these claims.

Clarity:
The authors' writings are clear on their approach and process. They described in great detail most of the work in the paper, and also provided a good amount of references on the literature and the methods that they use from other works.

Significance:
The results of the paper could be implemented in many downstream tasks in the research area, such as personalized chatbots, gaming, entertainment, etc. The method of identifying and modifying influential neurons in the LLMs could also be applied across broader range of use cases for LLMs outside of simulating personality traits.

**Weaknesses:**

The paper lacks some experiments and evaluation results that could help substantiate their validity. The authors did not describe how they know the dataset they constructed is a "good" dataset appropriate for this task other than a reference to one other paper. They also did not explain why using ChatGPT in the data generation and response evaluation process, although they did include a brief mention of some citations in the related work section.

**Questions:**

The authors should elaborate on the evaluation of the quality of the constructed PersonalityBench dataset. They could also include some experiments to compare training time and efficiency of their method compared to prompting-based and training-based methods; currently, they only compare their performance results rated by ChatGPT. On the same line of thought, they should give some explanations as to why ChatGPT would be a valid evaluation mechanism to provide personality trait scores for the models; only some references for personality assessment using LLMs are provided.


In terms of the presentation of the paper, the authors made some errors that could benefit from a slight revision. There are many empty spaces on page 16 that seem out of place. Furthermore, table 14 and table 15 are duplicates.

---

> ### Author Response · Authors · 2024-11-21
> **Response to Weakness 1**
>
> >The paper lacks some experiments and evaluation results that could help substantiate their validity.
>
> Thank you for pointing out the shortcomings in our experiments. We have conducted generative induction and multi-choice induction experiments in the paper along with compatibility experiments, ablation experiments, distribution analysis, and case studies in the original paper. To further demonstrate the validity of our approach, we have conducted more experiments during the response period:
> 1. Ablation studies on layers (Appendix A.5, Figure 5).
>
> 2. Additional case studies (Appendix D).
>
> 3. Human evaluation on PersonalityBench (Appendix B).
>
> 4. Experimental results on GSM8K, IFEVAL, and CommonsenseQA show that the model's general ability remains largely unaffected after NPTI (Appendix A.4).
>
> 5. The ActAdd baseline (Section 4.1, Reviewer WdT7's response under Weakness 2).
>
> 6. Consistency evaluation demonstrating a high correlation between ChatGPT-based automatic evaluation and human evaluation (Reviewer GZht's response under Weakness 4), as well as the inter-annotator agreement to demonstrate the high reliability of the human evaluation results (Reviewer WdT7's response under Weakness 4).
>
> If you believe we still lack any critical experiments, please feel free to let us know.

---

> > ### Author Response · Authors · 2024-11-21
> > **Response to Weakness 2&Question 1**
> >
> > >The authors did not describe how they know the dataset they constructed is a "good" dataset appropriate for this task other than a reference to one other paper.
> >
> > Thank you for highlighting the requirements of human evaluation for our datasets. Following your advice, we add a supplementary human evaluation process and provide further explanations of the dataset as follows.
> >
> > Our PersonalityBench consists of a personal description and a situational question. We select 20 questions per personality trait from the training and test sets, respectively, along with 10 descriptions per trait (5 positive, 5 negative), resulting in 50 descriptions and 400 questions (200 for training and 200 for test).
> >
> > We first **conduct a training process** for all human judges to provide clear guidelines and examples to ensure consistent evaluations. They need to evaluate whether situational questions can result in different responses for people with opposite traits. To further ensure the reliability of the participants, we also **require them to briefly describe how individuals with opposite personality traits would respond to the question**. Additionally, if they judge a particular data point as invalid, they **provide an explanation for their judgment**. As for personality biases, we also require the judges to check whether the question leads to a specific bias (e.g., emotional, moral, or personality) and write corresponding reasons. We also assess the quality of personal descriptions by requiring judges to check whether the description can reflect certain traits. The detailed instructions for annotators can be found in Tables 22 and 23 in Appendix D.
> >
> > Five Chinese undergraduate students, all CET-6 certified with strong English proficiency, participate in the assessments. The annotation takes place in a controlled lab environment over two consecutive days, with participants working from 10 a.m. to 7 p.m. (excluding breaks) for about 7 hours daily. Each data annotation takes approximately 2 minutes, and participants receive 0.5 dollars per data.
> > We list the ratio of valid and unbiased instances for each judge as follows:
> > \\begin{array}{||c|cccccc|c||}
> > \\hline
> > \\textrm{Trait} & \\textrm{Measure} & \\textrm{1} & \\textrm{2} & \\textrm{3} & \\textrm{4} & \\textrm{5} & \\textrm{Maj@1} \\\\
> > \\hline
> > \\textrm{Agreeableness} & \\textrm{Valid} & 0.98 & 0.85 & 0.95 & 0.98 & 0.93 & 0.99 \\\\
> >  & \\textrm{Unbiased} & 0.94 & 0.99 & 0.86 & 0.90 & 0.89 & 0.98 \\\\
> > \\hline
> > \\textrm{Conscientiousness} & \\textrm{Valid} & 0.98 & 0.84 & 0.91 & 0.96 & 0.85 & 0.98 \\\\
> >  & \\textrm{Unbiased} & 0.99 & 0.94 & 0.89 & 0.80 & 0.83 & 0.99 \\\\
> > \\hline
> > \\textrm{Extraversion} & \\textrm{Valid} & 0.94 & 0.83 & 0.85 & 0.95 & 0.85 & 0.90 \\\\
> >  & \\textrm{Unbiased} & 0.84 & 0.93 & 0.94 & 0.84 & 0.86 & 0.94 \\\\
> > \\hline
> > \\textrm{Neuroticism} & \\textrm{Valid} & 0.94 & 0.83 & 0.84 & 0.98 & 0.95 & 0.98 \\\\
> >  & \\textrm{Unbiased} & 0.95 & 0.98 & 0.99 & 0.93 & 0.91 & 0.98 \\\\
> > \\hline
> > \\textrm{Openness} & \\textrm{Valid} & 0.88 & 0.81 & 0.81 & 0.94 & 0.91 & 0.95 \\\\
> >  & \\textrm{Unbiased} & 0.93 & 0.96 & 0.96 & 0.88 & 0.86 & 0.99 \\\\
> > \\hline
> > \\textrm{Personality Description} & \\textrm{Valid} & 0.86 & 0.78 & 0.82 & 0.90 & 0.86 & 0.90 \\\\
> >  & \\textrm{Unbiased} & 1.00 & 0.98 & 1.00 & 1.00 & 1.00 & 1.00 \\\\
> > \\hline
> > \\end{array}
> >
> > We can find that most of our instances are of high quality, and nearly all the instances are recognized by at least 3 judges (maj@1). We have added the results in Appendix B in the revised version.

---

> ### Author Response · Authors · 2024-11-21
> **Response to Weakness 3&Question 3**
>
> >They also did not explain why using ChatGPT in the data generation and response evaluation process, although they did include a brief mention of some citations in the related work section.
>
> Thank you for pointing out the lack of explanation of why using ChatGPT in the data generation and response.
>
> 1. **Use ChatGPT in the data generation**: ChatGPT has been **widely used** to generate synthetic data across various domains, significantly enhancing model training and performance [1] [2] [3]. Moreover, leveraging ChatGPT has been shown to be an effective approach for creating synthetic datasets that retain the statistical properties of real data, making it particularly valuable in scenarios where data scarcity is a challenge [4]. Given the current lack of open-ended scenario-based questions for personality testing, we leverage the 300 descriptions of personality traits from the IPIP-NEO-300 to construct such questions. However, due to the sheer volume of questions, **manual creation is not feasible**. Therefore, we opt for the efficient use of ChatGPT to automate data generation. Moreover, to ensure the quality of the generated data, we implement a series of measures and conduct a human evaluation of the dataset to ensure its usefulness.
>
> 2. **Use ChatGPT in the response evaluation process**
>
> (1) **Efficiency**:
> Nowadays, ChatGPT-based automatic evaluation is widely applied in many tasks, reflecting its broad acceptance as a reliable tool for automated assessment [5] [6]. We believe that using ChatGPT-based automatic evaluation can **significantly reduce the cost of human annotation**.
>
> (2) **Consistency and Reproducibility**:
> LLMs can offer reproducible results. Once trained and fine-tuned for specific evaluation tasks, LLMs can **apply the same evaluation criteria uniformly**, ensuring that their judgments are repeatable over time and across different datasets. This leads to more objective and trustable results, which is widely applied in existing benchmarks, such as Arena-Hard and AlignBench.
>
> (3) **Validity**:
> We calculate the correlation between ChatGPT-based automatic evaluation and our human evaluation, which has shown a high correlation. Specifically, we calculate the consistent ratio of the ten partially ordered pairs of ChatGPT-based evaluation and five human judges. The results are shown below.
>
> \\begin{array}{||c|c|c|c|c||}
> \\hline
> \\textrm{1} & \\textrm{2} & \\textrm{3} & \\textrm{4} & \\textrm{5} \\\\
> \\hline
> 0.94 & 0.86 & 0.85 & 0.93 & 0.92 \\\\
> \\hline
> \\end{array}
>
> From the results above, we can see ChatGPT's evaluation has high correlation with human judgers. This indicates a high level of agreement between ChatGPT-based evaluation and human preferences, demonstrating the validity of our method.
>
> References:
>
> [1] Enhancing Chat Language Models by Scaling High-quality Instructional Conversations
>
> [2] Self-Instruct: Aligning Language Models with Self-Generated Instructions
>
> [3] WizardLM: Empowering Large Language Models to Follow Complex Instructions
>
> [4] Generative AI for Synthetic Data Generation: Methods, Challenges and the Future
>
> [5] Judging LLM-as-a-Judge with MT-Bench and Chatbot Arena
>
> [6] G-Eval: NLG Evaluation using GPT-4 with Better Human Alignment

---

> > ### Author Response · Authors · 2024-11-21
> > **Response to  Question 2**
> >
> > >They could also include some experiments to compare training time and efficiency of their method compared to prompting-based and training-based methods
> >
> > Thank you for emphasizing the lack of efficiency comparison in our work. We now supplement the following experiments.
> >
> > \\begin{array}{||c|c|c|c|c|c||}
> > \\hline
> > & \\textrm{PAS} & \\textrm{SFT} & \\textrm{P2} & \\textrm{Simple Prompt} & \\textrm{NPTI} \\\\
> > \\hline
> > \\textrm{Train/Search} & 0.28 & 0.26 & 0 & 0 & 0.11 \\\\
> > \\hline
> > \\textrm{Test} & 2.02 & 0.08 & 0.08 & 0.08 & 0.09 \\\\
> > \\hline
> > \\end{array}
> >
> > Our experiments are conducted on a single A800 GPU. For the train/identification stage, we calculate the average time spent per question (in seconds) across 36,000 questions. For the test stage, we measured the average time taken to evaluate each response (in seconds) across 450 questions.
> > From the results above, we can observe that the **efficiency of neuron search in NPTI surpasses the training efficiency of SFT**. Additionally, during the inference phase, our neuron induction method **only leads to minor time addition** compared with baseline methods. We have added the results in Appendix A.2 in the revised version.

---

> > > ### Author Response · Authors · 2024-11-21
> > > **Response to  Question 4**
> > >
> > > >In terms of the presentation of the paper, the authors made some errors that could benefit from a slight revision. There are many empty spaces on page 16 that seem out of place. Furthermore, table 14 and table 15 are duplicates.
> > >
> > > Thank you very much for pointing out the writing and formatting errors in our paper. In the latest version, we have used a grammar-checking tool for refinement and made corrections to some writing errors.

---

> ### Author Response · Authors · 2024-11-24
>
> Dear Reviewer hLAp,
>
> As the discussion is nearing its conclusion, we kindly ask if we have adequately addressed your concerns. If there are any additional questions or points requiring clarification, we would be more than happy to address them promptly.
>
> We deeply value your feedback and look forward to hearing from you soon.

---

> ### Author Response · Authors · 2024-11-27
>
> Dear Reviewer hLAp,
>
> Thank you once again for your valuable feedback, which has played a key role in improving our work. If we have successfully addressed all your concerns, we kindly ask for your consideration in updating the scores. Should you have any further comments or suggestions, we would be more than happy to discuss them with you.

---

> ### Author Response · Authors · 2024-11-30
>
> Dear reviewer hLAp,
>
> Thank you once again for your valuable feedback, which has greatly contributed to the improvement of our work. If we have adequately addressed all of your concerns, we would be grateful for your consideration in updating the review scores. Should you have any additional comments or suggestions, we would be more than happy to discuss them further.

---

> ### Author Response · Authors · 2024-12-02
>
> Dear Reviewer hLAp,
>
> As we near the conclusion of this discussion, we would like to ensure that we have addressed all of your concerns adequately. If there are any remaining questions or aspects that require further clarification, please feel free to let us know, and we would be happy to provide additional information.
>
> We greatly appreciate your insightful feedback and, if all concerns have been resolved, kindly ask for your consideration in updating the scores. We look forward to hearing from you soon.

---

### Official Review · Reviewer_GZht · 2024-11-03

**Soundness:** 3
**Presentation:** 3
**Contribution:** 3
**Rating:** 8
**Confidence:** 3

**Summary:**

This paper proposes “NPTI” (Neuron-based Personality Trait Induction), a method for inducing personality traits in LLMs by identifying and manipulating specific neurons associated with the Big Five personality traits (openness, conscientiousness, extroversion, agreeableness, neuroticism). The authors introduce three main contributions: (1) PERSONALITYBENCH, a dataset of 180,000 instances designed for evaluating and inducing specific personality traits in LLMs through generative tasks; (2) an identification method for personality-related neurons based on activation differences between opposing trait aspects; and (3) a targeted neuron manipulation technique to shift personality expressions, offering an efficient alternative to model fine-tuning.

**Strengths:**

- Precision in Personality Control: The neuron manipulation approach offers fine-grained control over specific traits in LLMs, presenting a compelling alternative to traditional prompt-based or full-model fine-tuning methods.
- PERSONALITYBENCH Dataset: PERSONALITYBENCH is designed for generative personality evaluation rather than traditional multiple-choice, providing a rich framework for testing personality consistency in LLMs.
- Method Efficiency: NPTI is more efficient than training-based personality manipulation methods, preserving the base model’s parameters and computational requirements.
- Cross-Model Testing: The evaluation on multiple models (LLaMA, Mistral, Gemma) illustrates NPTI’s compatibility across architectures, strengthening its practical significance.

**Weaknesses:**

- Ambiguity in Neuron Selection Thresholds: The neuron selection relies on a fixed 10% activation difference threshold (δ), but the justification for this choice is unclear. How this threshold affects neuron specificity and personality expression is not fully explored. Varying δ and showing results on a range of thresholds would provide insight into its impact on model behavior.
- Evaluation of Personality Consistency: While PERSONALITYBENCH includes 180,000 instances, it is unclear if the dataset covers enough variation to reflect nuanced personality traits in diverse contexts. More details on the dataset’s situational diversity would clarify its robustness in testing consistency.
- Stability and Fluency Trade-off: The paper briefly mentions a trade-off between personality expression and response fluency as γ varies but lacks a systematic evaluation of how different γ values affect output quality across traits. A more detailed analysis of this trade-off, potentially with fluency ratings per personality trait, would make the choice of γ more transparent and reproducible.
- Evaluation Metric Transparency: Automatic evaluations rely on ChatGPT’s trait and fluency ratings but lack details on how consistency and reliability are ensured across responses. Clearer definitions of these evaluation metrics and consistency checks would increase confidence in the method’s performance.

**Questions:**

- Can the authors clarify the selection of the 10% activation threshold (δ) for neuron identification? What rationale led to this value, and how does it impact the specificity and reliability of selected neurons?
- How does the chosen γ value impact the trade-off between personality intensity and fluency? Could the authors provide an empirical breakdown of how personality traits change with varying γ values and corresponding fluency scores?
- Could the authors explain the consistency checks in the automatic evaluation setup? Since ChatGPT rates personality intensity and fluency, are there mechanisms to ensure consistency across varied prompts?

---

> ### Author Response · Authors · 2024-11-21
> **Response to Weakness 1 & Question 1**
>
> >Can the authors clarify the selection of the 10% activation threshold (δ) for neuron identification? What rationale led to this value, and how does it impact the specificity and reliability of selected neurons?
>
> Thank you for highlighting the importance of conducting an ablation study on δ. However, we have already performed this experiment in Section 4.3.2, which is across all five personality dimensions and includes a brief analysis of the results. Specifically, **as the threshold decreases and more neurons are selected, the personality score gradually increases while the fluency score declines**. We chose a threshold of 10% because it represents the point where the **fluency score stabilizes while the personality score remains relatively high**.
>
> If you have any other questions, feel free to let us know!

---

> > ### Author Response · Authors · 2024-11-21
> > **Response to Weakness 2**
> >
> > >Evaluation of Personality Consistency: While PERSONALITYBENCH includes 180,000 instances, it is unclear if the dataset covers enough variation to reflect nuanced personality traits in diverse contexts. More details on the dataset’s situational diversity would clarify its robustness in testing consistency.
> >
> > Thank you for your concerns about the variation in our dataset. We employ many methods to guarantee the diversity of our dataset.
> > 1. **30 main topics and corresponding subtopics**: We incorporate 30 common real-world topics from UltraChat, such as technology, environment, and arts. For each topic and personality behavior, we set the temperature to 1 and top-p to 0.9, prompting GPT to generate 10 questions at a time, with each question focusing on a different subtopic of the main topic. This process is repeated twice, resulting in the complete dataset.
> >
> > 2. **300 personality facet descriptions from IPIP-NEO-300 questionnaire**: We utilize 300 personality behavior descriptions from the IPIP-NEO-300 questionnaire to generate the dataset. For each personality behavior description, we create 20 data samples tailored to different subtopics within each of the 30 topics. This ensures that the generated data captures diverse contexts and subtopic variations for every personality behavior.
> >
> > 3. **Validation of Dataset Diversity**: We compute the Distinct-4 [1] of our instances in each trait, which is a wide-spread metric to evaluate the text diversity. From the results in the table, we can see a high diversity score in each trait. That is, around half of the 4-grams are unique among 36,000 instances.
> >
> > \\begin{array}{||c|c|c|c|c||}
> > \\hline
> > \\textrm{Agreeableness} & \\textrm{Conscientiousness} & \\textrm{Extraversion} & \\textrm{Neuroticism} & \\textrm{Openness} \\\\
> > \\hline
> > 0.53 & 0.54 & 0.46 & 0.48 & 0.55 \\\\
> > \\hline
> > \\end{array}
> >
> > References:
> >
> > [1] A Diversity-Promoting Objective Function for Neural Conversation Models

---

> > > ### Author Response · Authors · 2024-11-21
> > > **Response to Weakness 3 &Question 2**
> > >
> > > >How does the chosen γ value impact the trade-off between personality intensity and fluency? Could the authors provide an empirical breakdown of how personality traits change with varying γ values and corresponding fluency scores?
> > >
> > > Thank you for highlighting the importance of conducting an ablation study on  γ. However, we have already performed this experiment in Section 4.3.2, which is across all five personality dimensions and includes a brief analysis of the results.
> > >
> > > Specifically, as the γ value increases–indicating a greater extent of neuron modification–**the personality score rises while the fluency score declines**. This pattern holds true for both positive and negative dimensions of the five personalities. To maintain readability, we choose the γ that **yields the highest personality score among the points where the fluency score does not fall below the fluency score of prompt-based methods rounded to one decimal place**. This helps avoid a significant gap in fluency compared to prompt-based methods.
> > >
> > > If you have any other questions, feel free to let us know!

---

> > > > ### Author Response · Authors · 2024-11-21
> > > > **Response to Weakness 4 & Question 3**
> > > >
> > > > >Evaluation Metric Transparency: Automatic evaluations rely on ChatGPT’s trait and fluency ratings but lack details on how consistency and reliability are ensured across responses. Clearer definitions of these evaluation metrics and consistency checks would increase confidence in the method’s performance.
> > > >
> > > > Thank you for pointing out the lack of clarity on how consistency and reliability are ensured in our evaluation metrics.
> > > > 1. **Clear Definitions of Automatic Evaluation**
> > > >
> > > > (1) **Automatic evaluation of personality traits**: We have provided the instruction in Table 12 in Appendix A. We instruct ChatGPT to remain as objective as possible while evaluating personality traits. Specific facets of the given trait are provided, ensuring that the scoring is based on well-defined criteria rather than being arbitrary. To enhance interpretability, ChatGPT is required to **provide a brief explanation** before assigning a score, demonstrating the reasoning behind the evaluation. Furthermore, the scoring is constrained to a 1-5 scale, where higher scores indicate better alignment with the specified trait.
> > > >
> > > > (2) **Automatic evaluation fluency**: We have provided the instruction in Table 13 in Appendix A. We instruct ChatGPT to evaluate the fluency of responses with a strict focus on **how smoothly and naturally thoughts and emotions are conveyed**, explicitly excluding considerations like language formality or response length. To enhance interpretability, ChatGPT is required to provide a concise explanation of no more than 30 words before assigning a score, ensuring the reasoning behind the evaluation is clear. The scoring follows a 1-5 scale, with higher scores indicating better fluency, seamlessness, and clarity of expression.
> > > >
> > > > 2. **Consistency check**
> > > >
> > > > We calculate the correlation between ChatGPT evaluation and our human evaluation, which has shown a high correlation. Specifically, we calculate the consistent ratio of the ten partially ordered pairs of ChatGPT ranking and human ranking. The results are shown below.
> > > >
> > > > \\begin{array}{||c|c|c|c|c||}
> > > > \\hline
> > > > \\textrm{1} & \\textrm{2} & \\textrm{3} & \\textrm{4} & \\textrm{5} \\\\
> > > > \\hline
> > > > 0.94 & 0.86 & 0.85 & 0.93 & 0.92 \\\\
> > > > \\hline
> > > > \\end{array}
> > > >
> > > > From the results above, we can see ChatGPT evaluation has **high correlation** with human judges. This indicates a high level of agreement between ChatGPT evaluation and human preferences, demonstrating the consistency of our evaluation.

---

> ### Author Response · Authors · 2024-11-24
>
> Dear Reviewer GZht,
>
> As the discussion is nearing its conclusion, we kindly ask if we have adequately addressed your concerns. If there are any additional questions or points requiring clarification, we would be more than happy to address them promptly.
>
> We deeply value your feedback and look forward to hearing from you soon.

---

> ### Author Response · Authors · 2024-11-27
>
> Dear Reviewer GZht,
>
> Thank you once again for your valuable feedback, which has played a key role in improving our work. If we have successfully addressed all your concerns, we kindly ask for your consideration in updating the scores. Should you have any further comments or suggestions, we would be more than happy to discuss them with you.

---

> > ### Comment · Reviewer_GZht · 2024-11-27
> > **Update of the score.**
> >
> > Dear Author,
> >
> > Thanks for the replies. I have updated my scores.

---

> > > ### Author Response · Authors · 2024-11-27
> > > **Heartfelt Gratitude**
> > >
> > > Dear Reviewer GZht,
> > >
> > > Thank you so much for your kind feedback and for updating the scores. We deeply appreciate your recognition of our work—it truly means a lot to us. Your thoughtful comments have been incredibly helpful, and we are grateful for the time and effort you dedicated to reviewing our paper.
> > >
> > > As this season of gratitude approaches, we would like to wish you a very joyful and peaceful Thanksgiving. May your holiday be filled with warmth, happiness, and cherished moments with your loved ones.

---

### Official Review · Reviewer_WdT7 · 2024-11-04

**Soundness:** 3
**Presentation:** 2
**Contribution:** 3
**Rating:** 6
**Confidence:** 4

**Summary:**

The paper presents a neuron manipulation approach to inducing Big Five personality traits in LLMs without retraining or altering model parameters. The paper introduces PersonalityBench, a large-scale generative dataset based on real-world scenarios and grounded in the Big Five personality traits. By identifying specific personality-related neurons, the work enables neuron manipulation that allows the models to exhibit targeted personality traits.

**Strengths:**

The paper proposes a new neuron manipulation method that effectively induces diferent personality traits in LLMs by providing two prompts with contrasting personality levels. The approach also demonstrates the feasibility through three different evaluations.

The paper is clearly written, with diagrams that are easy to follow and overall pleasing to look at.

**Weaknesses:**

1. The situational questions used for automatic evaluation are generated by ChatGPT, yet they have not been verified by human reviewers. Additionally, details on how SocialIQA data contributed to constructing these questions are not provided. This raises concerns that SocialIQA may already embed certain personality biases, which could compromise the validity of the automatic evaluation.

2. A similar work outlined here: https://arxiv.org/abs/2308.10248 appears to closely resemble the neuron manipulation method proposed in this paper. Including this work as a baseline would provide a more comprehensive comparison.

3. The paper only provides a general scaled rating for generation fluency, and the scores in Table 17 seem uniformly close to 10 across all methods. This might reflect overly lenient evaluation criteria, rather than true performance. A single GPT-based metric is also not reliable enough. To address this, the authors should evaluate the model on broader benchmarks, e.g., assessing general instruction-following capabilities and factual knowledge retention to ensure the approach does not degrade overall performance.

4. There are no details about the human evaluation setup and results. For example, the paper does not mention the expertise levels of the evaluators, nor does it discuss inter-annotator agreement. Given that subtle differences between personality expressions might be challenging for laypeople to detect, it would help to specify if raters are experts or had specialized training. Additionally, the current task setup requires ranking five items, which is more complex than rating them individually. See question (4).

5. The IPIP-NEO-120 and IPIP-NEO-300 assessments overlap, which weakens the credibility of the multiple-choice evaluation. This issue is evident in Table 3, where the PAS method (based on IPIP-NEO-300) also yields a low score, suggesting that overlapping data could affect the accuracy of the personality alignment test.

**Questions:**

1. Could you clarify the motivation and the validity of the automatic evaluation method? For example, is there a strong correlation between the benchmark results and the IPIP-NEO-300 personality test?
2. What data was used for the supervised fine-tuning process? Was it drawn from PersonalityBench, or were other datasets involved?
3. It appears that neuron manipulations are applied across all layers. Have you conducted any ablation studies focusing on manipulating neurons in specific layers only, and if so, how does this impact performance?
4 A rating-based evaluation might provide clearer insights into model performance. Could you elaborate on why a rating-based approach was not chosen?

---

> ### Author Response · Authors · 2024-11-21
> **Response to Weakness 1**
>
> >The situational questions used for automatic evaluation are generated by ChatGPT, yet they have not been verified by human reviewers. Additionally, details on how SocialIQA data contributed to constructing these questions are not provided. This raises concerns that SocialIQA may already embed certain personality biases, which could compromise the validity of the automatic evaluation.
>
> Thank you for highlighting the requirements of human evaluation for our datasets. Following your advice, we add a supplementary human evaluation process and provide further explanations of the dataset as follows.
>
> 1. **Human Evaluation for PersonalityBench**:
> Our PersonalityBench consists of a personal description and a situational question. We select 20 questions per personality trait from the training and test sets, respectively, along with 10 descriptions per trait (5 positive, 5 negative), resulting in 50 descriptions and 400 questions (200 for training and 200 for test).
> We first conduct a training process for all human judges to provide clear guidelines and examples to ensure consistent evaluations. They need to evaluate whether situational questions can result in different responses for people with opposite traits. To further ensure the reliability of the participants, we also require them to briefly describe how individuals with opposite personality traits would respond to the question. Additionally, if they judge a particular data point as invalid, they **provide an explanation** for their judgment. As for personality biases, we also require the judges to check whether the question leads to a specific bias (e.g., emotional, moral, or personality) and write corresponding reasons. We also assess the quality of personal descriptions by requiring judges to check whether the description can reflect certain traits. The detailed instructions for annotators can be found in Tables 22 and 23 in Appendix D.
> Five Chinese undergraduate students, all CET-6 certified with strong English proficiency, participate in the assessments. The annotation takes place in a controlled lab environment over two consecutive days, with participants working from 10 a.m. to 7 p.m. (excluding breaks) for about 7 hours daily. Each data annotation takes approximately 2 minutes, and participants receive 0.5 dollars per data.
> We list the ratio of valid and unbiased instances for each judge as follows:
> \\begin{array}{||c|cccccc|c||}
> \\hline
> \\textrm{Trait} & \\textrm{Measure} & \\textrm{1} & \\textrm{2} & \\textrm{3} & \\textrm{4} & \\textrm{5} & \\textrm{Maj@1} \\\\
> \\hline
> \\textrm{Agreeableness} & \\textrm{Valid} & 0.98 & 0.85 & 0.95 & 0.98 & 0.93 & 0.99 \\\\
>  & \\textrm{Unbiased} & 0.94 & 0.99 & 0.86 & 0.90 & 0.89 & 0.98 \\\\
> \\hline
> \\textrm{Conscientiousness} & \\textrm{Valid} & 0.98 & 0.84 & 0.91 & 0.96 & 0.85 & 0.98 \\\\
>  & \\textrm{Unbiased} & 0.99 & 0.94 & 0.89 & 0.80 & 0.83 & 0.99 \\\\
> \\hline
> \\textrm{Extraversion} & \\textrm{Valid} & 0.94 & 0.83 & 0.85 & 0.95 & 0.85 & 0.90 \\\\
>  & \\textrm{Unbiased} & 0.84 & 0.93 & 0.94 & 0.84 & 0.86 & 0.94 \\\\
> \\hline
> \\textrm{Neuroticism} & \\textrm{Valid} & 0.94 & 0.83 & 0.84 & 0.98 & 0.95 & 0.98 \\\\
>  & \\textrm{Unbiased} & 0.95 & 0.98 & 0.99 & 0.93 & 0.91 & 0.98 \\\\
> \\hline
> \\textrm{Openness} & \\textrm{Valid} & 0.88 & 0.81 & 0.81 & 0.94 & 0.91 & 0.95 \\\\
>  & \\textrm{Unbiased} & 0.93 & 0.96 & 0.96 & 0.88 & 0.86 & 0.99 \\\\
> \\hline
> \\textrm{Personality Description} & \\textrm{Valid} & 0.86 & 0.78 & 0.82 & 0.90 & 0.86 & 0.90 \\\\
>  & \\textrm{Unbiased} & 1.00 & 0.98 & 1.00 & 1.00 & 1.00 & 1.00 \\\\
> \\hline
> \\end{array}
> We can find that most of our instances are of high-quality and nearly all the instances are recognized by at least 3 judges (maj@1). We have added the results in Appendix B in the revised version.
>
> 2.  **Construction Test Set Using SocialIQA**: We leverage SocialIQA to assist in constructing our test set with several efforts to make it more reliable. Specifically, we first prompt ChatGPT to rewrite SocialIQA questions in the second person and select those that reflect a specific personality trait. Next, we instruct ChatGPT to enhance the scenarios underlying these questions by introducing complexities, such as dilemmas, while ensuring they remain consistent with the original context. For revision, we apply the same debiasing prompt used for the training set to review potential biases and make necessary improvements. Finally, our co-authors manually filter questions that are free of bias and effectively capture personality traits. In total, we identify 459 questions, with an average of 92 questions per personality dimension.
> Thanks for your concerns and we will add these details and results to the final version.

---

> ### Author Response · Authors · 2024-11-21
> **Response to Weakness 2**
>
> >A similar work outlined here: https://arxiv.org/abs/2308.10248 appears to closely resemble the neuron manipulation method proposed in this paper. Including this work as a baseline would provide a more comprehensive comparison.
>
> Thank you for carefully reviewing our approach and conducting a thorough comparison. Following your advice, we further analyze and compare the differences and effectiveness of the two methods (i.e., ActAdd and NPTI (ours)).
>
> 1. **Comparison of the Two Methods**
>
> Our NPTI method has several differences:
>
> (1) NPTI modifies FFN layers at the neuron level, while ActAdd applies changes to the residual stream values.
>
> (2) NPTI adjusts all layers, whereas ActAdd modifies a single layer, which requires hyperparameter tuning using grid search.
>
> (3) NPTI identifies personality-related neurons using **activation probability differences**, with larger probability differences indicating more important neurons and thus greater modification magnitudes. To account for each neuron's activation patterns, we use its **95th percentile activation** as a constraint. In contrast, ActAdd directly applies the activation value difference between opposing concepts without such considerations.
>
> 2. **Experimental Results**
>
> We use GitHub code from its official repository to implement ActAdd and adopt grid search to determine the optimal hyperparameters. We have tested two prompting approaches: (1) using a pair of antonyms (e.g., "extraverted" and "introverted"), and (2) using detailed descriptions of opposing personality traits. Ultimately, we select the second approach based on the results.
> The comparison of personality score and fluency score is as follows:
>
> Personality score:
> \\begin{array}{||c|c|c|c|c|c||}
> \\hline
> \\textrm{Method} & \textrm{Agreeableness} & \textrm{Conscientiousness} & \textrm{Extraversion} & \textrm{Neuroticism} & \\textrm{Openness} \\\\
> \\hline
> \textrm{NPTI} & 9.64  \\pm 0.49 & 9.25  \\pm 0.66 & 9.86  \\pm 0.14 & 9.92  \\pm 0.07 & 8.50  \\pm 1.08 \\\\
> \textrm{ActAdd} & 8.20  \\pm 2.90 & 6.61  \\pm 2.75 & 8.84  \\pm 1.44 & 8.90  \\pm 1.78 & 8.52  \\pm 1.83 \\\\
> \\hline
> \\end{array}
>
>
> Fluency score:
> \\[
> \\begin{array}{||c|c|c|c|c|c||}
> \\hline
> \\textrm{Method} & \\textrm{Agreeableness} & \\textrm{Conscientiousness} & \\textrm{Extraversion} & \\textrm{Neuroticism} & \\textrm{Openness} \\\\
> \\hline
> \\textrm{NPTI} & 9.72\\pm  0.23 & 9.96\\pm  0.04 & 9.88\\pm  0.11 & 9.90\\ pm 0.09 & 9.83\\pm  0.18 \\\\
> \\textrm{AddAct} & 8.69\\pm  1.58 & 8.92\\pm  1.31 & 8.80\\pm  1.71 & 9.14\\ pm 1.34 & 8.79\\pm  1.78 \\\\
> \\hline
> \\end{array}
> \\]
> Under each personality trait, the first number represents the score, while the second number represents the variance. From the table, we can see that NPTI outperforms ActAdd in both effectiveness and stability across agreeableness, conscientiousness, extraversion, and neuroticism. For openness, while the personality scores are slightly lower, NPTI demonstrates greater stability. Regarding fluency, NPTI significantly surpasses ActAdd in fluency scores. We have added the results in Section 4.1 in the revised version.
>
> Besides scores, when comparing the generated content, we find that **ActAdd tends to briefly recall and restate the description prompt rather than understanding and answering the questions**, which may be a severe issue and an explanation for lower fluency scores. From the following case, it can be found that the answer is the second person and just repeat the given description.
>
> **Description**: You are an extroverted person who thrives in social settings. Your energy and enthusiasm are infectious, and you are often the life of the party. You enjoy meeting new people and are comfortable taking the lead in conversations. Your assertiveness and sociability make you a natural networker, always eager to build new connections and friendships.
>
> **Question**: When you find yourself in a social gathering, how likely are you to initiate a long conversation about your personal experiences with someone you've just met?
>
> **Answer from ActAdd**: You are the best party hostess ever! You have an infectious enthusiasm for meeting  new people and making new friends. Your outgoing personality is infectious, and people love being around you because of the energy and excitement that comes with being around the life of the party!

---

> > ### Author Response · Authors · 2024-11-21
> > **Response to Weakness 3**
> >
> > >The paper only provides a general scaled rating for generation fluency, and the scores in Table 17 seem uniformly close to 10 across all methods. This might reflect overly lenient evaluation criteria, rather than true performance. A single GPT-based metric is also not reliable enough. To address this, the authors should evaluate the model on broader benchmarks, e.g., assessing general instruction-following capabilities and factual knowledge retention to ensure the approach does not degrade overall performance.
> >
> > Thank you for your constructive feedback regarding the lack of evaluation of general capabilities. Follow your advice, we add experiments as follows:
> >
> > We select GSM8K, IFEval, and CommonsenseQA benchmarks to test the LLaMA-3-8B-Instruct model's general instruction-following capabilities and factual knowledge retention after applying NPTI. For GSM8K and CommonsenseQA, we use the configurations reported in the official LLaMA documentation, while for IFEval, we adopt a 0-shot setting. The model's performance is listed as follows:
> >
> > \\begin{array}{|c|c|c|c|c|}
> > \\hline
> > \\textrm{} & \\textrm{GSM8K} & \\textrm{IFEval (loose)} & \\textrm{IFEval (strict)} & \\textrm{CommonsenseQA} \\\\
> > \\hline
> > \\textrm{Base} & 77.9 & 81.5 & 75.8 & 76.5 \\\\
> > \\hline
> > \\textrm{Agreeableness+} & 76.8\\, (\\downarrow\\, 1.1) & 78.1\\, (\\downarrow\\, 3.4) & 72.1\\, (\\downarrow\\, 3.7) & 76.0\\, (\\downarrow\\, 0.5) \\\\
> > \\hline
> > \\textrm{Agreeableness-} & 74.7\\, (\\downarrow\\, 3.2) & 79.0\\, (\\downarrow\\, 2.5) & 71.1\\, (\\downarrow\\, 4.7) & 74.0\\, (\\downarrow\\, 2.5) \\\\
> > \\hline
> > \\textrm{Conscientiousness+} & 78.2\\, (\\uparrow\\, 0.3) & 81.9\\, (\\uparrow\\, 0.4) & 76.2\\, (\\uparrow\\, 0.4) & 77.1\\, (\\uparrow\\, 0.6) \\\\
> > \\hline
> > \\textrm{Conscientiousness-} & 76.0\\, (\\downarrow\\, 1.9) & 80.6\\, (\\downarrow\\, 0.9) & 75.2\\, (\\downarrow\\, 0.6) & 75.4\\, (\\downarrow\\, 1.1) \\\\
> > \\hline
> > \\textrm{Extraversion+} & 76.7\\, (\\downarrow\\, 1.2) & 76.1\\, (\\downarrow\\, 5.4) & 70.6\\, (\\downarrow\\, 5.2) & 75.7\\, (\\downarrow\\, 0.8) \\\\
> > \\hline
> > \\textrm{Extraversion-} & 74.5\\, (\\downarrow\\, 3.4) & 81.3\\, (\\downarrow\\, 0.2) & 73.6\\, (\\downarrow\\, 2.2) & 75.1\\, (\\downarrow\\, 1.4) \\\\
> > \\hline
> > \\textrm{Neuroticism+} & 75.5\\, (\\downarrow\\, 2.4) & 77.8\\, (\\downarrow\\, 3.7) & 71.1\\, (\\downarrow\\, 4.7) & 69.5\\, (\\downarrow\\, 7.0) \\\\
> > \\hline
> > \\textrm{Neuroticism-} & 77.4\\, (\\downarrow\\, 0.5) & 80.2\\, (\\downarrow\\, 1.3) & 73.2\\, (\\downarrow\\, 2.6) & 75.8\\, (\\downarrow\\, 0.7) \\\\
> > \\hline
> > \\textrm{Openness+} & 77.1\\, (\\downarrow\\, 0.8) & 80.5\\, (\\downarrow\\, 1.0) & 73.6\\, (\\downarrow\\, 2.2) & 76.3\\, (\\downarrow\\, 0.2) \\\\
> > \\hline
> > \\textrm{Openness-} & 73.8\\, (\\downarrow\\, 4.1) & 80.1\\, (\\downarrow\\, 1.4) & 75.9\\, (\\uparrow\\, 0.1) & 75.3\\, (\\downarrow\\, 1.2) \\\\
> > \\hline
> > \\end{array}
> >
> > From the table, we can find that most of the results decline slightly, which can verify our effective identification of neurons since we adjust one-tenth of neurons.
> >
> > Surprisingly, we observe that activating neurons associated with the positive aspect of conscientiousness leads to improvement in all tasks. By examining the responses, we find that the model **provides detailed explanations** for the reasons behind each answer.
> >
> > In contrast, activating neurons associated with the positive aspect of neuroticism leads to the most noticeable decline across benchmarks. Upon reviewing the model-generated responses, we observe that the model tends to **exhibit increased anxiety and lack of confidence** in its explanations, which subsequently impacts the correctness of its answers.
> >
> > These findings are consistent with our intuition and further verify the effectiveness of our NPTI method. We have added the results in Appendix A.4 in the revised version.

---

> > > ### Author Response · Authors · 2024-11-21
> > > **Response to Weakness 4**
> > >
> > > >There are no details about the human evaluation setup and results. For example, the paper does not mention the expertise levels of the evaluators, nor does it discuss inter-annotator agreement. Given that subtle differences between personality expressions might be challenging for laypeople to detect, it would help to specify if raters are experts or had specialized training. Additionally, the current task setup requires ranking five items, which is more complex than rating them individually. See question (4).
> > > Thank you for pointing out the lack of details on human evaluation. We'd like to supplement our details of Table 2 as follows:
> > >
> > > 1. **The Quality of Evaluators**: The five evaluators we choose are Chinese undergraduate students, all CET-6 certified with strong English proficiency.
> > >
> > > 2. **Training Session Prior to the Assessment**: Before starting the annotation process, we conduct a focused training session for the five evaluators. Specifically, we first have the evaluators complete the IPIP-NEO-300 test to familiarize themselves with the descriptive traits of each personality dimension. Next, we provide detailed explanations of the Big Five personality traits, along with their positive and negative characteristics, to aid their understanding. To ensure clarity on the evaluation rules, we also guide them through five ranking exercises as hands-on practice.
> > >
> > > 3. **Inter-annotator Agreement**:
> > > We calculate the pairwise agreement among five participants for rankings of 5 responses to 200 questions. As for one question-answer pair, two judges have their ranks in five responses. We calculate the consistent ratio of the ten partially ordered pairs. The results are as follows:
> > > \\begin{array}{|c|c|c|c|c|c|}
> > > \\hline
> > > \\textrm{Evaluator} & 1 & 2 & 3 & 4 & 5 \\\\
> > > \\hline
> > > 1 & 1 & 0.81 & 0.82 & 0.87 & 0.74 \\\\
> > > 2 & 0.81 & 1 & 0.65 & 0.73 & 0.67 \\\\
> > > 3 & 0.82 & 0.65 & 1 & 0.74 & 0.67 \\\\
> > > 4 & 0.87 & 0.73 & 0.74 & 1 & 0.78 \\\\
> > > 5 & 0.74 & 0.67 & 0.67 & 0.78 & 1 \\\\
> > > \\hline
> > > \\end{array}
> > >
> > > The pairwise agreement probabilities among the five participants range from 0.67 to 0.87, indicating **a relatively high level of consistency**. We have added the results in Appendix A.3 in the revised version.
> > >
> > > 4.**The Reason for Choosing Ranking Rather than Rating**: We present our answer in **Response to Question 4**.

---

> > > > ### Author Response · Authors · 2024-11-21
> > > > **Response to Weakness 5**
> > > >
> > > > >The IPIP-NEO-120 and IPIP-NEO-300 assessments overlap, which weakens the credibility of the multiple-choice evaluation. This issue is evident in Table 3, where the PAS method (based on IPIP-NEO-300) also yields a low score, suggesting that overlapping data could affect the accuracy of the personality alignment test.
> > > >
> > > > Thank you for pointing out the error we made. We review the code from the PAS paper and confirm that their data does not have an overlap issue. Additionally, we remove the duplicate questions from the IPIP-NEO-300 and retest our method. The updated results are as follows:
> > > >
> > > > \\[
> > > > \\begin{array}{||c|c|c|c|c|c|c||}
> > > > \\hline
> > > > \\textrm{Method} & \\textrm{Agreeableness} & \\textrm{Conscientiousness} & \\textrm{Extraversion} & \\textrm{Neuroticism} & \\textrm{Openness} & \\textrm{Total} \\\\
> > > > \\hline
> > > > \\textrm{PAS} & 0.94 & 0.91 & 0.86 & 0.98 & 0.72 & 4.41 \\\\
> > > > \\textrm{NPTI} & 0.90 & 0.90 & 0.86 & 0.90 & 0.86 & 4.42 \\\\
> > > > \\hline
> > > > \\end{array}
> > > > \\]

---

> > > > > ### Author Response · Authors · 2024-11-21
> > > > > **Response to Question 1**
> > > > >
> > > > > >Could you clarify the motivation and the validity of the automatic evaluation method? For example, is there a strong correlation between the benchmark results and the IPIP-NEO-300 personality test?
> > > > >
> > > > > Thank you for questions. We will explain why we use these evaluation benchmarks and the motivation and validity why we use ChatGPT for automatic evaluation in LLMs' generative capabilities.
> > > > >
> > > > > 1. **Overall Design of Our Evaluation**
> > > > >
> > > > > We utilize both generative evaluation (PersonalityBench) and multiple-choice evaluation (IPIP-NEO) to assess LLMs' personality induction abilities from different dimensions. We choose challenging generative evaluation based on PersonalityBench since it **shares a similar scenario when LLMs are applied in downstream tasks**. Most existing personality assessment methods primarily rely on questionnaire-based multiple-choice formats, which may lead to evaluation bias or even inaccuracies. Furthermore, to align with multiple-choice evaluation and demonstrate the generality of our method, we also add IPIP-NEO as an evaluation and fairly compare with baseline method PAS.
> > > > >
> > > > > 2. **Motivation of ChatGPT Automatic Evaluation**
> > > > >
> > > > > (1) Efficiency: Nowadays, ChatGPT-based automatic evaluation is **widely applied** in many tasks, reflecting its broad acceptance as a reliable tool for automated assessment [1] [2]. We believe that using ChatGPT-based automatic evaluation can significantly reduce the cost of human annotation. It automates the evaluation process, offering a highly efficient alternative to traditional human-dependent methods, while maintaining scalability and adaptability in various contexts.
> > > > >
> > > > > (2) Consistency and Reproducibility: LLMs can offer reproducible results. Once trained and fine-tuned for specific evaluation tasks, LLMs can apply the same evaluation criteria uniformly, ensuring that their judgments are repeatable over time and across different datasets. This leads to more objective and trustable results, which is widely applied in existing benchmarks, such as Arena-Hard and AlignBench. In contrast, human evaluations often suffer from subjective bias, fatigue, and variability in judgment. This subjectivity makes it difficult to reproduce assessments reliably across different evaluators.
> > > > >
> > > > > 3. **Validity of ChatGPT-Based Automatic Evaluation**
> > > > >
> > > > > We calculate the correlation between ChatGPT-based automatic evaluation and our human evaluation, which shows a high correlation. Specifically, we calculate the consistent ratio of the ten partially ordered pairs of ChatGPT-based evaluation and five human judges. The results are shown below.
> > > > >
> > > > > \\begin{array}{||c|c|c|c|c||}
> > > > > \\hline
> > > > > \\textrm{1} & \\textrm{2} & \\textrm{3} & \\textrm{4} & \\textrm{5} \\\\
> > > > > \\hline
> > > > > 0.94 & 0.86 & 0.85 & 0.93 & 0.92 \\\\
> > > > > \\hline
> > > > > \\end{array}
> > > > >
> > > > > From the results above, we can see ChatGPT's evaluation has **high correlation with human judges**. This indicates a high level of agreement between ChatGPT-based evaluation and human preferences, demonstrating the validity of our method.

---

> > > > > > ### Author Response · Authors · 2024-11-21
> > > > > > **Response to Question 2**
> > > > > >
> > > > > > >What data was used for the supervised fine-tuning process? Was it drawn from PersonalityBench, or were other datasets involved?
> > > > > >
> > > > > > To ensure consistency and control variables with our NPTI method, the questions used for SFT are the same as PersonalityBench, while the answers are **generated by the model itself** during the process of identifying personality-related neurons (i.e., responses produced under detailed prompts designed to elicit answers aligned with specific personality traits).

---

> > > > > > > ### Author Response · Authors · 2024-11-21
> > > > > > > **Response to Question 3**
> > > > > > >
> > > > > > > >It appears that neuron manipulations are applied across all layers. Have you conducted any ablation studies focusing on manipulating neurons in specific layers only, and if so, how does this impact performance?
> > > > > > >
> > > > > > > Thank you for emphasizing the importance of conducting ablation studies on layers. In response, we conduct layer ablation experiments on traits of agreeableness and neuroticism. We keep the \gamma values consistent with those in the original paper and explore the effects of activating neurons in only a single layer and neurons across five consecutive layers. The resulting changes in personality scores are shown in Figure 5 in Appendix B.
> > > > > > >
> > > > > > > In the figure, the green line represents the scores by activating neurons across all layers. The red line serves as the threshold for determining the effectiveness of the method (i.e., not activating neurons). Scores above the red line indicate that the relevant neurons are effective in inducing personality traits in the model, whereas scores below suggest ineffectiveness. From the experimental results, we observe that **neurons in the middle and bottom layers play a more critical role** in shaping the model’s personality, with those **in the middle layers being particularly impactful**. The results also show that **relying on several layers cannot lead to optimal performance**, and determining which layers to use is also time-consuming. This is why we activate neurons across all layers.
> > > > > > >
> > > > > > > Thanks for your advice and we will add these results in the final version.

---

> > > > > > > > ### Author Response · Authors · 2024-11-21
> > > > > > > > **Response to Question 4**
> > > > > > > >
> > > > > > > > >A rating-based evaluation might provide clearer insights into model performance. Could you elaborate on why a rating-based approach was not chosen?
> > > > > > > >
> > > > > > > > We choose a ranking-based evaluation over a rating-based approach for several reasons:
> > > > > > > > 1. **Reduction of Subjective Bias**: Research suggests that ranking methods can reduce variability caused by subjective biases in absolute scoring [1]. When participants assign scores, their interpretation of the scoring scale may vary significantly, leading to inconsistencies. Ranking, on the other hand, focuses on relative preferences, making it more robust to such biases.
> > > > > > > > 2. **Minimizing Calibration Bias**: Rating-based evaluations are susceptible to calibration effects, where participants may rate all responses similarly high or low, reducing the discriminatory power of the scores [2]. Ranking forces evaluators to distinguish between responses, offering clearer insights into performance differences.
> > > > > > > >
> > > > > > > > To conclude, by adopting a ranking-based approach, the evaluation maintains a clear and consistent methodology, reducing noise and improving reliability compared to a rating-based system. This choice aligns with existing LLM benchmarks, such as Chatbot Arena and Arena-Hard.
> > > > > > > >
> > > > > > > > References:
> > > > > > > >
> > > > > > > > [1] Multiple Criteria Decision Analysis: An Integrated Approach
> > > > > > > >
> > > > > > > > [2] Fast and accurate inference of Plackett-Luce models

---

> ### Author Response · Authors · 2024-11-24
>
> Dear Reviewer WdT7,
>
> As the discussion is nearing its conclusion, we kindly ask if we have adequately addressed your concerns. If there are any additional questions or points requiring clarification, we would be more than happy to address them promptly.
>
> We deeply value your feedback and look forward to hearing from you soon.

---

> ### Author Response · Authors · 2024-11-27
>
> Dear Reviewer WdT7,
>
> Thank you once again for your valuable feedback, which has played a key role in improving our work. If we have successfully addressed all your concerns, we kindly ask for your consideration in updating the scores. Should you have any further comments or suggestions, we would be more than happy to discuss them with you.

---

> ### Author Response · Authors · 2024-11-30
>
> Dear reviewer WdT7,
>
> Thank you once again for your valuable feedback, which has greatly contributed to the improvement of our work. If we have adequately addressed all of your concerns, we would be grateful for your consideration in updating the review scores. Should you have any additional comments or suggestions, we would be more than happy to discuss them further.

---

> > ### Comment · Reviewer_WdT7 · 2024-12-02
> >
> > Thank you for the authors' detailed responses. I have carefully reviewed them and updated my score accordingly.

---

> ### Author Response · Authors · 2024-12-02
> **Thank you**
>
> Dear reviewer WdT7,
>
> Thank you for your kind words and for taking the time to review our responses. We truly appreciate your thoughtful feedback and updated score. Your detailed comments have been very helpful in refining our work, and we are grateful for your careful consideration！

---

### Author Response · Authors · 2024-12-04
**New Revision of our Paper**

We would like to express our sincere gratitude to the three reviewers for their valuable and constructive comments. In response to their suggestions, we have revised our paper accordingly. Below, we outline the key changes made:

- Consistency and Reliability Check in Evaluation Results(Reviewers WdT7, GZht, hLAp): We calculate the consistency ratio among the five participants in Appendix A.3, demonstrating a high inter-annotator agreement and confirming the reliability of our manual ranking. Moreover, we calculate the correlation between ChatGPT evaluation and our human evaluation, which shows a high correlation.

- Evaluation on PersonalityBench (Reviewer WdT7, GZht, hLAp): We add a supplementary human evaluation on PersonalityBench in Appendix B, improving the dataset's reliability. Additionally, we provide further details to demonstrate that it captures sufficient variation to reflect nuanced personality traits across different contexts, ensuring its robustness in testing consistency.

- Ablation Studies on Layers (Reviewer WdT7): We conduct ablation experiments on the layers in Appendix A.5 and find that the middle layers are the most important. However, relying on multiple layers doesn't yield optimal performance, confirming the effectiveness of adjusting all layer parameters.

- Evaluation on Broader Benchmarks (Reviewer WdT7): We evaluate the model using NPTI on broader benchmarks such as GSM8K and IFEval in Appendix A.4, demonstrating that the approach does not significantly degrade general capabilities.

- Comparison with ActAdd (Reviewer WdT7): We incorporate ActAdd as an additional baseline method, and the results demonstrate that NPTI remains the top performer.

- IPIP-NEO-300 Duplicate Removal (Reviewer WdT7): We remove the duplicate questions from the IPIP-NEO-300 and retest our method. The updated results show that our approach remains competitive.

- Training and Inference Efficiency Comparison (Reviewer hLAp): We compare the training and inference efficiency of our method in Appendix A.2. We observe that the efficiency of neuron search in NPTI exceeds that of SFT training. Additionally, during inference, our neuron induction method adds only a slight time overhead compared to baseline methods.

- Additional Case Studies(Reviewer hLAp): We include additional case studies in Appendix C, demonstrating that our method can integrate different personality traits and outperforms simple prompt-based induction.

Finally, we once again express our sincere gratitude to the reviewers for their valuable feedback and suggestions. We believe these revisions and additions further enhance the quality and contribution of our paper.

---

### Meta-Review · Area_Chair_spn8 · 2024-12-21

**Metareview:**

This paper proposes a neuron manipulation method for inducing Big Five personality traits in large language models (LLMs) without the need for retraining or modifying model parameters. It also introduces PersonalityBench, a large-scale generative dataset grounded in real-world scenarios and aligned with the Big Five personality traits. By identifying personality-specific neurons, the approach enables targeted neuron manipulation, allowing models to exhibit desired personality traits.

Pros:
1. The paper is clearly written and easy to follow.
2. The paper introduces a neuron manipulation method to induce varying personality traits in LLMs using prompts with contrasting personality levels, validated through three evaluations.
3. PersonalityBench dataset is designed for generative personality evaluation rather than traditional multiple-choice, providing a rich framework for testing personality consistency in LLMs.

Cons:
1. Situational questions for automatic evaluation were generated by ChatGPT without human verification, and the influence of SocialIQA data remains unclear. Potential biases in SocialIQA could compromise evaluation validity.
2. A similar neuron manipulation method described in prior work (arXiv:2308.10248) is not included as a baseline, limiting the comprehensiveness of the comparison.
3. Fluency scores (Table 17) appear uniformly high, suggesting lenient criteria rather than robust performance. Reliance on a single GPT-based metric is insufficient, and broader benchmarks (e.g., instruction-following, factual knowledge retention) are needed to ensure the approach does not degrade overall capabilities.
4. The paper lacks details on human evaluation, including evaluator expertise, inter-annotator agreement, and task complexity. Subtle personality expression differences may require trained experts for accurate assessment.

This seems a borderline paper to me. There are several key weaknesses identified by the reviewers on the evaluation. The authors have addressed some of them during the rebuttal. I feel the PersonalityBench dataset might be a good contribution which could potentially benefit for future research.

**Additional Comments On Reviewer Discussion:**

This seems a borderline paper to me. There are several key weaknesses identified by the reviewers on the evaluation. The authors have addressed some of them during the rebuttal. Overall, I feel the PersonalityBench dataset might be a good contribution which could potentially benefit for future research.

---

### Decision · Program_Chairs · 2025-01-22

Accept (Poster)